# CONVERGENT GRAPH SOLVERS

**Junyoung Park, Jinhyun Choo & Jinkyoo Park**
KAIST, Daejeon, South Korea
`{junyoungpark,jinhyun.choo,jinkyoo.park}@kaist.ac.kr`

## ABSTRACT

We propose the convergent graph solver **(CGS)**[1], a deep learning method that learns iterative mappings to predict the properties of a graph system at its stationary state (fixed point) *with guaranteed convergence*. The forward propagation of CGS proceeds in three steps: (1) constructing the input-dependent linear contracting iterative maps, (2) computing the fixed points of the iterative maps, and (3) decoding the fixed points to estimate the properties. The contractivity of the constructed linear maps guarantees the existence and uniqueness of the fixed points following the Banach fixed point theorem. To train CGS efficiently, we also derive a tractable analytical expression for its gradient by leveraging the implicit function theorem. We evaluate the performance of CGS by applying it to various network-analytic and graph benchmark problems. The results indicate that CGS has competitive capabilities for predicting the stationary properties of graph systems, irrespective of whether the target systems are linear or non-linear. CGS also shows high performance for graph classification problems where the existence or the meaning of a fixed point is hard to be clearly defined, which highlights the potential of CGS as a general graph neural network architecture.

## 1 INTRODUCTION

Our world is replete with networked systems, where their overall properties emerge from complex interactions among the system entities. Such networked systems attain their unique properties from their stationary states; hence, finding these stationary properties is a common goal for many problems that arise in the science and engineering field. Examples of such problems include the minimization of the molecule's potential energy that finds the stationary positions of the atoms to compute the potential energy of the molecule (Moloi & Ali, 2005), the PageRank algorithm that finds the stationary importance of online web pages to compute recommendation scores (Brin & Page, 1998), and the network analysis of fluid flow in porous media that finds the stationary pressures in pore networks to compute the macroscopic properties of the media (Gostick et al., 2016). In these network-analytic problems, the network is often represented as a graph, and the stationary states are often computed through a problem-specific iterative method derived analytically from the prior knowledge of the target system. By applying the iterative map repeatedly, these methods compute the stationary states (i.e., fixed points) and the associated properties of the target system. However, analytically deriving such iterative maps for highly complex systems typically requires tremendous efforts and time.

Instead of deriving/designing the problem-specific iterative methods, researchers employ deep learning approaches to *learn* the iterative methods (Dai et al., 2018; Gu et al., 2020; Hsieh et al., 2019; Huang et al., 2020). These approaches learn an iterative map directly without using domain-specific knowledge, but using only the input and output data. By applying the learned iterative map, these approaches (approximately) compute the fixed points and predict the properties of a target system. Graph Neural Networks (GNNs) have been widely used to construct iterative maps (Dai et al., 2018; Gu et al., 2020; Scarselli et al., 2008; Alet et al., 2019). However, when applying this approach, the existence of the fixed points is seldom guaranteed. As a result, the number of iterative steps is often required to be specified, as a hyperparameter, to ensure the termination of the iterations. This may cause premature termination or inefficient backward propagation if the number of iterations is inadequately small or too large.

---

[1]The code is available at https://github.com/Junyoungpark/CGS.

In this study, we propose a convergent graph solver (CGS), a deep learning method that can predict the solution of a target graph analytical problem using only the input and output data, and without requiring the prior knowledge of existing solvers or intermediate solutions. The forward propagation of CGS is designed to proceed in the following three steps:

- **Constructing the input-dependent linear-contracting iterative maps.** CGS uses the input graph, which dictates the specification of the target network-analytic problem, to construct a set of linear contracting maps. This procedure formulates/set up the internal problem to be solved by considering the problem conditions and contexts (i.e., boundary conditions or initial conditions in PDE domains – the physical network problems). Furthermore, the input-dependent linear map can produce any size of transition map flexibly depending on the input size graph; thus helping the trained model to generalize over unseen problems with different sizes (size transferability).

- **Computing the fixed points via iterative methods**. CGS constructs a set of linear contracting maps, each of which is guaranteed to have a unique fixed point that embeds the important features for conducting various end tasks. Thus, CGS computes the unique solutions of the constructed linear maps via iterative methods (or direct inversion) with convergence guarantee.

- **Decoding the fixed points to estimate the properties**. By using a separate decoder architecture, we compute the fixed points in the latent space while expecting them to be an effective representation that can improve the predictive performance of the model. This enables CGS to be used not only for finding the real fixed points (or its transportation) if they exist, but also for computing the "virtual fixed point" as a representation learning method in general prediction tasks.

The parameters of CGS are optimized with the gradient computed based on the implicit function theorem, which requires $\mathcal{O}(1)$ memory usage when computing the gradient along with the iterative steps. CGS is different from the other studies that solve the constrained optimization (with convergence guarantee) (Gu et al., 2020; Scarselli et al., 2008; Tiezzi et al., 2020) in that it does not impose any restriction when optimizing the parameters. Instead, CGS is inherently structured to have the unique fixed points owing to the uses of the linear map. Note that the structural restriction is only in the form of an iterative map; we can flexibly generate the coefficients of the linear map using any network (i.e., GNN) and utilize multiple linear maps to boost the representability.

We evaluate the performance of CGS using two types of paradigmatic network-analytic problems: physical diffusion in networks and the Markov decision process, where the true labels indeed exist and can be computed analytically using linear and non-linear iterative methods, respectively. We also employ CGS to solve various graph classification benchmark tasks to show that CGS can serve as a general (implicit) layer like other GNN networks when conducting typical graph classification tasks. In these experiments, we seek to compute a *virtual fixed point* that can serve as the best hidden representation of the input when predicting the output. The results show that CGS can be (1) an effective solver for graph network problems or (2) an effective general computational layer for processing graph-structured data.

## 2 RELATED WORK

**Convergent neural models.** Previous studies that have attempted to achieve the convergence property of neural network embedding (e.g., hidden vectors of MLP and hidden states of recurrent neural networks) can be grouped into two categories: *soft* and *hard* approaches. Soft approaches typically attain the desired convergence properties by augmenting the loss functions (Erichson et al., 2019; Miyato et al., 2018). Although the soft approaches are network-architecture agnostic, these methods cannot guarantee the convergence of the learned mappings. On the other hand, hard approaches seek to guarantee the convergence of the iterative maps by restricting their parameters in certain ranges (Gu et al., 2020; Tiezzi et al., 2020; Miller & Hardt, 2018; Kolter & Manek, 2019). This is achieved by projecting the parameters of the models into stable regions. However, such projection may lead to non-optimal performances since it is performed after the gradient update, i.e., the projection is disentangled from training objective.

**Implicit deep models.** The forward propagation of CGS, which solves fixed point iterations, is closely related to deep implicit models. Instead of defining the computational procedures (e.g., the depth of layer in neural network) explicitly, these models use implicit layers, which accept input-dependent equations and compute the solutions of the input equations, for the forward propagation.

For instance, neural ordinary differential equations (NODE) (Chen et al., 2018; Massaroli et al., 2020) solve ODE until the solver tolerance is satisfied or the integration domain is covered, the optimization layers (Gould et al., 2016; Amos & Kolter, 2017) solve the optimization problem until the duality gap converges, and the fixed point models (Bai et al., 2019; Winston & Kolter, 2020) solve the network-generated fixed point iterations until some numerical solver satisfy the convergence condition. Implicit models use these (intermediate) solutions to conduct various end tasks (e.g., regressions, or classifications). In this way, implicit models can impose desired *behavioral characteristics* into layers as inductive bias and hence, often show superior parameter/memory efficiency and predictability.

**Fixed points of graph convolution.** Methods that find the fixed points of graph convolutions have been suggested in various contexts of graph-related tasks. Several works have utilized GNN along with RNN-like connections to (approximately) find the fixed points of graph convolutions (Liao et al., 2018; Dai et al., 2018; Li et al., 2015; Scarselli et al., 2008). Some have suggested to constrain the parameter space of GNN so that the trained GNN becomes a non-expansive map, thus producing the fixed point (Gu et al., 2020; Tiezzi et al., 2020). Others have proposed to apply an additional GNN layer on the embedded graph and penalize the difference between the output of the additional GNN layer and the embedded graph to guide the GNN to find the fixed points (Scarselli et al., 2008; Yang et al., 2021). It has been shown that regularizing GNN to find its fixed points improves the predictive performance (Tiezzi et al., 2020; Yang et al., 2021).

**Comparison between CGS and existing approaches.** Combining the ideas of (1) computing/using the fixed points of the graph convolution (as a representation learning) and (2) utilizing the implicit differentiation of the model (as a training method) has been proposed by numerous studies (Scarselli et al., 2008; Liao et al., 2018; Johnson et al., 2020; Dai et al., 2018; Bai et al., 2019; Gallicchio & Micheli, 2020; Bai et al., 2019; Tiezzi et al., 2020; Gu et al., 2020). Majority of those studies *assume* that the convolution operators (iterative map) induce a convergent sequence of the (hidden) representation; however, this assumption typically does not hold unless certain conditions hold for the convolution operator (iterative map). When neither the convergence nor uniqueness holds, it can possibly introduce biases in the gradient computed by implicit differentiation (Liao et al., 2018; Blondel et al., 2021). Hence, to impose convergence, some methods restrict the learned convolutions to be strictly contractive (i.e., the hidden solutions are convergent) by projecting the learned parameters into a certain region and solving the constraint training problems, respectively (Gu et al., 2020; Tiezzi et al., 2020). Unlike these methods restricting the parameters of the graph convolutions directly, CGS guarantees the convergence and the uniqueness of the fixed point by imposing the structural inductive bias on the iterative map, i.e., using the contractive linear map that has a unique fixed point, thus alleviating the need to solve the constrained parameter optimization (Tiezzi et al., 2020).

## 3 PROBLEM DESCRIPTION

The objective of many network-analytic problems can be described as:

*Find a solution vector $\boldsymbol{Y}^*$ from a graph $\mathcal{G}$ that represents the target network system.*

In this section, we briefly explain a general iterative scheme to compute $\boldsymbol{Y}^*$ from $\mathcal{G}$. The problem specification $\mathcal{G} = (\mathbb{V}, \mathbb{E})$ is a directed graph that is composed of a set of nodes $\mathbb{V}$ and a set of edges $\mathbb{E}$. We define the $i^{\text{th}}$ node as $v_i$ and the edge from $v_i$ to $v_j$ as $e_{ij}$. The general scheme of the iterative methods is given as follows:

$$\boldsymbol{H}^{[0]} = f(\mathcal{G}), \tag{1}$$

$$\boldsymbol{H}^{[n]} = \mathcal{T}(\boldsymbol{H}^{[n-1]}; \mathcal{G}), \qquad n = 1, 2, \dots \tag{2}$$

$$\boldsymbol{Y}^{[n]} = g(\boldsymbol{H}^{[n]}; \mathcal{G}), \qquad n = 0, 1, \dots \tag{3}$$

where $f(\cdot)$ is the problem-specific initialization scheme that transforms $\mathcal{G}$ into the initial hidden embedding $\boldsymbol{H}^{[0]}$, $\mathcal{T}$ is the problem-specific iterative map that updates the hidden embedding $\boldsymbol{H}^{[n]}$ from the previous embedding $\boldsymbol{H}^{[n-1]}$, and $g(\cdot)$ is the problem-specific decoding function that predicts the intermediate solution $\boldsymbol{Y}^{[n]}$.

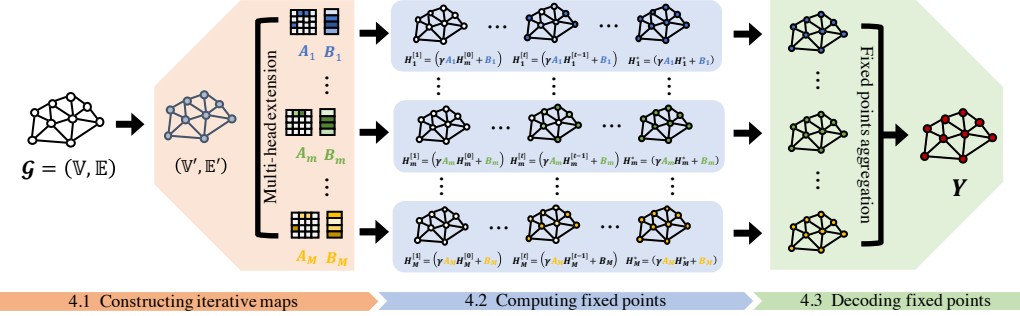

Figure 1: **Overview of forward propagation of CGS.** Given an input graph $\mathcal{G}$, the parameter-generating network $f_\theta$ constructs contracting linear transition maps $\mathcal{T}_\theta$. The fixed points $\boldsymbol{H}_m^*$ of $\mathcal{T}_\theta$ are then computed via matrix inversion. The fixed points are aggregated into $\boldsymbol{H}^*$, and then the decoder $g_\theta$ decodes $\boldsymbol{H}^*$ to produce $\boldsymbol{Y}^*$.

$\mathcal{T}$ is *designed* such that the fixed point iteration (equation 2) converges to the unique fixed point $\boldsymbol{H}^*$:

$$\lim_{n\to\infty} \boldsymbol{H}^{[n]} = \boldsymbol{H}^* \text{ s.t. } \boldsymbol{H}^* = \mathcal{T}(\boldsymbol{H}^*, \mathcal{G}) \tag{4}$$

The solution $\boldsymbol{Y}^*$ is then obtained by decoding $\boldsymbol{H}^*$, i.e., $g(\boldsymbol{H}^*; \mathcal{G}) \triangleq \boldsymbol{Y}^*$.

In many real-world network-analytic problems, we can obtain $\mathcal{G}$ and its corresponding solution $\boldsymbol{Y}^*$, but not $\mathcal{T}$, $\boldsymbol{H}^{[n]}$ and $\boldsymbol{Y}^{[n]}$. Therefore, we aim to learn a mapping $\mathcal{T}$ from $\mathcal{G}$ to $\boldsymbol{Y}^*$ without using $\mathcal{T}$, $\boldsymbol{H}^{[n]}$ and $\boldsymbol{Y}^{[n]}$.

### 3.1 EXAMPLE: GRAPH VALUE ITERATION

Let us consider a finite Markov decision process (MDP), whose goal is to find the state values through the iterative applications of the Bellman optimality backup operator (Bellman, 1954). We assume that the state transition is deterministic.

We define $\mathcal{G} = (\mathbb{V}, \mathbb{E})$, where $\mathbb{V}$ and $\mathbb{E}$ are the set of states and transitions of MDP respectively. $v_i$ corresponds to the $i^{\text{th}}$ state of the MDP, and $e_{ij}$ corresponds to the state transition from state $i$ to $j$. $e_{ij}$ exists only if the state transition from state $i$ to $j$ is allowed in the MDP. The features of $e_{ij}$ are the corresponding state transition rewards. The objective is to find the state values $V^*$. In this setting, the Bellman optimal backup operator $\mathcal{T}$ is defined as follows:

$$V_i^{[n]} = \mathcal{T}(V^{[n-1]}; \mathcal{G}) \triangleq \max_{j\in\mathcal{N}(i)} \left( r_{ij} + \alpha V_j^{[n-1]} \right) \tag{5}$$

where $V_i^{[n]}$ is the state value of $v_i$ estimated at $n$-th iteration, $\mathcal{N}(i)$ is the set of states that can be reached from the $i^{\text{th}}$ state via one step transition, $r_{ij}$ is the immediate reward related to $e_{ij}$, and $\alpha$ is the discount rate of the MDP.

In this graph value iteration (GVI) problem, the initial values $V^{[0]}$ are set as zeros (i.e., $f(\cdot) = \boldsymbol{0}$), then $\mathcal{T}$ is applied until $V^{[n]}$ converges, and $V^{[n]}$ is decoded with the identity mapping (i.e., $g(\cdot)$ is identity and $H^{[n]} \triangleq V^{[n]}$). We will show how CGS constructs the transition map $\mathcal{T}$ from the input graph and predicts the converged state values $V^*$ without using equation 5 in the following sections.

## 4 CONVERGENT GRAPH SOLVERS

We propose CGS that predicts the solution $\boldsymbol{Y}^*$ from the given input graph $\mathcal{G}$ in three steps: (1) constructing linear iterative maps $\mathcal{T}_\theta$ from $\mathcal{G}$, (2) computing the unique fixed points $\boldsymbol{H}^*$ of $\mathcal{T}_\theta$ via iterative methods, and (3) decoding $\boldsymbol{H}^*$ to produce $\boldsymbol{Y}^*$, as shown in Figure 1.

## 4.1 Constructing linear iterative maps

CGS first constructs an input-dependent contracting linear map $\mathcal{T}_\theta(\cdot\,;\mathcal{G})$ such that the repeated application of $\mathcal{T}_\theta(\cdot\,;\mathcal{G})$ *always* produces the unique fixed point $\boldsymbol{H}^*$ (i.e. $\lim_{n\to\infty}\boldsymbol{H}^{[n]} = \boldsymbol{H}^*$) that embeds the essential characteristics of the input graph $\mathcal{G}$ for conducting end tasks. In other words, CGS learns to construct iterative maps that are tailored to each input graph $\mathcal{G}$, from which the unique fixed points of the target system are *guaranteed* to be computed and used for conducting end tasks.

To impose the contraction property on the linear map, CGS utilizes the following iterative map:

$$\mathcal{T}_\theta(\boldsymbol{H}^{[n]};\mathcal{G}) \triangleq \gamma\boldsymbol{A}_\theta(\mathcal{G})\boldsymbol{H}^{[n]} + \boldsymbol{B}_\theta(\mathcal{G}) \tag{6}$$

where $\gamma$ is the contraction factor, $\boldsymbol{A}_\theta(\mathcal{G}) \in \mathbb{R}^{p\times p}$ is the input-dependent transition parameter, $\boldsymbol{B}_\theta(\mathcal{G}) \in \mathbb{R}^p$ is the input-dependent bias parameter, and $p$ is the number of nodes in graph.

To construct an input-dependent iterative map that preserves the structural constraints, which is required to guarantee the existence and uniqueness of a fixed point, we employ GNN-based parameter-generating network $f_\theta(\cdot)$. The parameter generation procedure for $\mathcal{T}_\theta$ starts by encoding $\mathcal{G} = (\mathbb{V},\mathbb{E})$:

$$\mathbb{V}', \mathbb{E}' = f_\theta(\mathbb{V},\mathbb{E}) \tag{7}$$

where $\mathbb{V}'$ and $\mathbb{E}'$ are the set of updated node embeddings $v_i' \in \mathbb{R}^q$ and edge embeddings $e_{ij}' \in \mathbb{R}$ respectively. CGS then constructs $\boldsymbol{A}_\theta(\mathcal{G})$ by computing the $(i,j)^{\text{th}}$ element of $\boldsymbol{A}_\theta(\mathcal{G})$ as follows:

$$[\boldsymbol{A}_\theta(\mathcal{G})]_{i,j} = \begin{cases} \frac{\sigma(e_{ij}')}{d(i)} & \text{if } e_{ij} \text{ exists,} \\ 0 & \text{otherwise.} \end{cases} \tag{8}$$

where $\sigma(x)$ is a differentiable bounded function that projects $x$ into the range $[0,1]$ (e.g., Sigmoid function), and $d(i)$ is the outward degree of $v_i$ (i.e., the number of outward edges of $v_i$). $\boldsymbol{B}_\theta(\mathcal{G})$ is simply constructed by vectorizing the updated node embeddings as $[\boldsymbol{B}_\theta(\mathcal{G})]_{i,:} = v_i'$.

**Theorem 1**. *The existence and uniqueness of $\boldsymbol{H}^*$ induced by $\mathcal{T}_\theta(\boldsymbol{H}^{[n]};\mathcal{G})$. The proposed scheme for constructing $\boldsymbol{A}_\theta(\mathcal{G})$, along with the bounded contraction factor $0 < \gamma < 1$, is sufficient for $\mathcal{T}_\theta(\boldsymbol{H}^{[n]};\mathcal{G})$ to be $\gamma$-contracting and, as a result, $\mathcal{T}_\theta$ has unique fixed point $\boldsymbol{H}^*$. Refer to Appendix A.1 for the proof.*

**Multi-head extension.** $\mathcal{T}_\theta$ can be considered as a graph convolution layer defined in equation 6. Thus, CGS can be easily extended to multiple convolutions in order to model a more complex iterative map. To achieve such *multi-head* extension with $M$ graph convolutions, one can design $f_\theta$ to produce a set of transition parameters $[\boldsymbol{A}_1,...,\boldsymbol{A}_m,...,\boldsymbol{A}_M]$ and a set of bias parameters $[\boldsymbol{B}_1,...,\boldsymbol{B}_m,...,\boldsymbol{B}_M]$ for $\mathcal{T}_m(\boldsymbol{H}_m^{[n]};\mathcal{G}) \triangleq \gamma\boldsymbol{A}_m\boldsymbol{H}_m^{[n]} + \boldsymbol{B}_m$ for $m = 1,...,M$.

## 4.2 Computing fixed points

The fixed point $\boldsymbol{H}_m^*$ of the constructed iterative map $\mathcal{T}_m(\boldsymbol{H}_m^{[n]};\mathcal{G}) \triangleq \gamma\boldsymbol{A}_m\boldsymbol{H}_m^{[n]} + \boldsymbol{B}_m$ satisfies $\boldsymbol{H}_m^* = \gamma\boldsymbol{A}_m\boldsymbol{H}_m^* + \boldsymbol{B}_m$ for $m = 1,...,M$. Due to the linearity, we can compute the fixed point of $\mathcal{T}_m$ via matrix inversion:

$$\boldsymbol{H}_m^* = (I - \gamma\boldsymbol{A}_m)^{-1}\boldsymbol{B}_m \tag{9}$$

where $I \in \mathbb{R}^{p\times p}$ is the identity matrix. The existence of $(I - \gamma\boldsymbol{A}_m)^{-1}$ is assured from the fact that $\mathcal{T}_m$ is contracting (see Appendix A.2). The matrix inversions can be found by applying various automatic differentiation tools while maintaining its differentiability. However, the computational complexity of the matrix inversion scales $\mathcal{O}(p^3)$, which can limit this approach from being scaled to large scale problems favorably.

To scale CGS to larger graph inputs, we compute the fixed point of $\mathcal{T}_m$ by repeatedly applying the iterative map $\mathcal{T}_m$ starting from an arbitrary initial hidden state $\boldsymbol{H}_m^{[0]} \in \mathbb{R}^p$ until the hidden embedding converges, i.e., $\lim_{n\to\infty}\boldsymbol{H}_m^{[n]} = \boldsymbol{H}_m^*$.

One can choose a way to compute the fixed point between inversion and iterative methods depending on the size of the transition matrix $\boldsymbol{A}_m$ and its sparsity because these factors can result in different computational speed and accuracy. In general, for small-sized problems, inversion methods can be favorable; while for large-sized problems, iterative methods are more efficient. (See Appendix D.6)

### 4.3 DECODING FIXED POINTS

The final step of CGS is to aggregate the fixed points of multiple iterative maps and decode the aggregated fixed points to produce $\boldsymbol{Y}^*$. The entire decoding step is given as follows:

$$\boldsymbol{H}^* = [\boldsymbol{H}_m^* || ... || \boldsymbol{H}_M^*] \tag{10}$$

$$\boldsymbol{Y}^* = g_\phi(\boldsymbol{H}^*; \mathcal{G}) \tag{11}$$

where $\boldsymbol{H}^*$ is the aggregated fixed points, $M$ is the number of heads, and $g_\phi(\cdot)$ is the decoder which is analogous to the decoding function of the network-analytic problems (equation 3).

## 5 TRAINING CGS

To train CGS with gradient descent, we need to calculate the partial derivatives of the scalar-valued loss $\mathcal{L}$ with respect to the parameters of $\mathcal{T}_\theta$. To do so, we express the partial derivatives using chain rule taking $\boldsymbol{H}^*$ as the intermediate variable

$$\frac{\partial \mathcal{L}}{\partial (\cdot)} = \frac{\partial \mathcal{L}}{\partial \boldsymbol{H}^*} \frac{\partial \boldsymbol{H}^*}{\partial (\cdot)} \tag{12}$$

where $(\cdot)$ denotes the parameters of $\boldsymbol{A}_\theta(\mathcal{G})$ or $\boldsymbol{B}_\theta(\mathcal{G})$. Here, $\frac{\partial \mathcal{L}}{\partial \boldsymbol{H}^*}$ is readily computable via an automatic differentiation package. However, computing $\frac{\partial \boldsymbol{H}^*}{\partial (\cdot)}$ is less straightforward since $\boldsymbol{H}^*$ and $(\cdot)$ are implicitly related via equation 6. One possible option to compute the partial derivatives is to backpropagate through the iteration steps.

Although this approach can be easily employed using most automatic differentiation tools, it entails extensive memory usage. Instead, exploiting the stationarity of $\boldsymbol{H}^*$, we can derive an analytical expression for the partial derivative using the implicit function theorem as follows:

$$\frac{\partial \boldsymbol{H}^*}{\partial (\cdot)} = -\left( \frac{\partial g(\boldsymbol{H}^*, \boldsymbol{A}, \boldsymbol{B})}{\partial \boldsymbol{H}^*} \right)^{-1} \frac{\partial g(\boldsymbol{H}^*, \boldsymbol{A}, \boldsymbol{B})}{\partial (\cdot)} \tag{13}$$

where $g(\boldsymbol{H}, \boldsymbol{A}, \boldsymbol{B}) = \boldsymbol{H} - (\gamma \boldsymbol{A} \boldsymbol{H} + \boldsymbol{B})$. Here, we omit the input-dependency of $\boldsymbol{A}$ and $\boldsymbol{B}$ for notational brevity. We compute the inverse terms via an iterative method. This option allows one to train CGS with constant memory consumption over the iterative steps. The derivation of the partial derivatives and the software implementation of equation 13 are provided in Appendix A.3 and B respectively.

## 6 EXPERIMENTS

We first evaluate the performance of CGS for two types of network-analytic problems: (1) the stationary state of physical diffusion in networks where the true solutions can be computed from linear iterative maps, and (2) the state values via GVI where the true solutions can be calculated from non-linear iterative maps. We then assess the capabilities of CGS as a general GNN layer by applying CGS to solve several graph property prediction benchmarks problems.

### 6.1 PHYSICAL DIFFUSION IN NETWORKS

Diffusion of fluid, heat, and other physical quantities are omnipresent in the science and engineering applications. Mathematically, physical diffusion in networks (e.g. pipe/pore networks) is often described by a graph. The stationary state of the graph can be expressed as

$$\sum_{j \in \mathcal{N}(i)} k_{ij}(p_i - p_j) = 0, \quad \forall v_i \in \mathbb{V} \setminus \partial(\mathbb{V}), \tag{14}$$

$$p_i = p_i^b, \quad \forall v_i \in \partial(\mathbb{V}), \tag{15}$$

where $p_i$ and $p_j$ are the potentials at $v_i$ and $v_j$ respectively, $k_{ij}$ is the conductance of the edge the connects the two nodes, and $p_i^b$ is the prescribed potentials at the boundary nodes that belong to the set $\partial(\mathbb{V})$. Equation 14 specializes to a particular diffusion problem according to how $p$ is prescribed (e.g. pressure for fluid flow, and temperature for heat transfer).

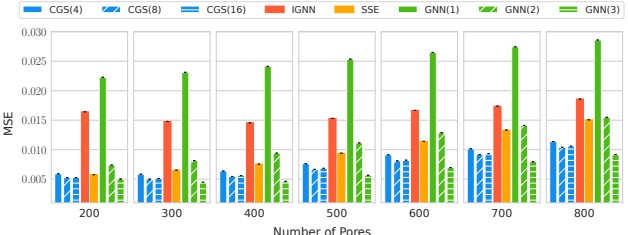
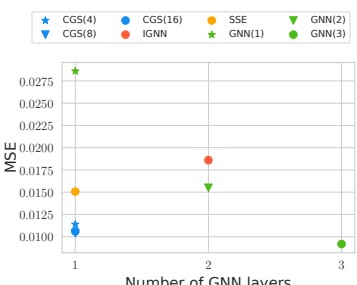

Figure 2: **Diffusion experiment results.** The x- and y-axis are the number of pores and the average MSE of the test graphs. The error bars indicates the standard error of predictions (measured from 500 instances per each size).

Figure 3: **Number of GNN layers vs. MSE ($n_s$ = 800).**

As an example of physical diffusion, we consider fluid flow in porous media – particularly, finding the fluid pressures of a pore network that is in equilibrium state (the solutions of equations 14 and 15). We model the pore network as a 3D graph whose nodes and edges correspond to pore chambers and throats respectively as shown in figure 4. We assume linear diffusion such that $p$ (i.e., $\boldsymbol{Y}^*$) can be computed using a *linear* iterative map (Gostick et al., 2016).

We employ CGS to predict the equilibrium pressures $\boldsymbol{Y}^*$ inside pore networks $\mathcal{G}$. The node features are the Cartesian coordinates, volume, diameter, and boundary indicator of its corresponding pore. The boundary pressure is also as a node feature if the node corresponds to a boundary pore. The edge features are the cylinder volume, diameter, and length of its corre-

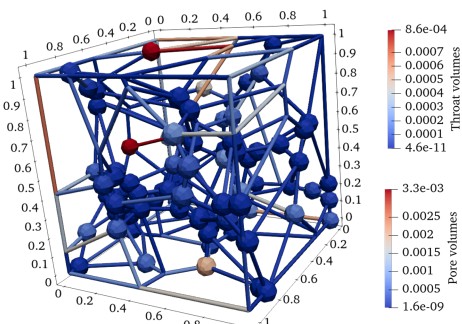

Figure 4: **Pore network graph**

sponding throat. We sample training graphs such that the graphs fit into 0.1 m³ cubes. The training graphs, which have 50–200 nodes, are then randomly generated (See Appendix C.1). We train CGS such that it minimizes the mean-squared error (MSE) between the predicted ones and $\boldsymbol{Y}^*$.

To investigate the effectiveness of the multi-head extension, we train CGS(4), CGS(8) and CGS(16), where CGS($m$) denotes CGS with $m$ heads. For the baseline models, we use implicit GNNs, IGNN, (Gu et al., 2020), SSE (Dai et al., 2018), and $n$-layer GNN models GNN($n$). IGNN and SSE find the fixed points in the forward propagation step. We utilize the same GNN architecture as the encoders for all baselines except for SSE. Please refer to Appendix C.1 for the details about the data generation, network architectures, and training schemes.

All CGS models show better generalization capabilities than the baselines in predicting $\boldsymbol{Y}^*$ as shown in Figure 2, even though all models show similar prediction errors during training (See Appendix C.1.2). CGSs with higher $m$ show superior prediction results for the test cases. This difference evinces that the use of multi-head extension (i.e., multiple linear iterative maps) is advantageous due to the increased expressivity. Also, when comparing CGS(8) and GNN(1), which utilize the same encoder architecture and thus has the same number of GNN layers as shown in Figure 3, CGS(8) shows better prediction performance. This is because the 1-hop aggregation cannot provide enough information to compute the equilibrium pressure. This result indicates that CGS successfully accommodates the long-range patterns in graphs without adopting additional graph convolution layers.

## 6.2 GRAPH VALUE ITERATION

We investigate the performance of CGS on graph value iteration (GVI) problems, where the iterative map (equation 5) is *non-linear* as explained in section 3.1. The goal of the experiments is to show that CGS can estimate the state values that are computed from the nonlinear iterative map accurately, even when using the set of learned linear iterative maps, as shown in figure 5.

Figure 5: **Solutions of GVI with CGS.** The balls represent the states of MDP and the ball colors show the prediction results and their corresponding targets. More details are described in the main text.

Table 1: **Graph Value Iteration results.** We report the average MAPE and policy prediction accuracies (in %) of different $n_s$ and $n_a$ combinations with 500 repeats per combination. All metrics are measured per graph.

| $n_s$ | 20 | | 50 | | 75 | | 100 | | #. params |
|---|---|---|---|---|---|---|---|---|---|
| $n_a$ | 5 | 10 | 10 | 15 | 10 | 15 | 10 | 15 | |
| SSE | $7.40 \pm 4.54$ $(0.75 \pm 0.11\,\%)$ | $5.98 \pm 3.30$ $(0.72 \pm 0.12\,\%)$ | $6.60 \pm 2.74$ $(0.70 \pm 0.08\,\%)$ | $97.52 \pm 0.11$ $(0.71 \pm 0.12\,\%)$ | $6.52 \pm 2.49$ $(0.69 \pm 0.06\,\%)$ | $97.51 \pm 0.06$ $(0.67 \pm 0.06\,\%)$ | $6.66 \pm 2.21$ $(0.68 \pm 0.06\,\%)$ | $97.50 \pm 0.05$ $(0.67 \pm 0.06\,\%)$ | 43,521 |
| IGNN | $13.87 \pm 4.69$ $(0.68 \pm 0.12\,\%)$ | $28.38 \pm 1.77$ $(0.63 \pm 0.13\,\%)$ | $28.13 \pm 1.40$ $(0.61 \pm 0.08\,\%)$ | $29.44 \pm 1.35$ $(0.62 \pm 0.13\,\%)$ | $28.21 \pm 1.29$ $(0.60 \pm 0.07\,\%)$ | $29.20 \pm 0.88$ $(0.60 \pm 0.07\,\%)$ | $28.00 \pm 1.15$ $(0.60 \pm 0.06\,\%)$ | $29.17 \pm 0.81$ $(0.60 \pm 0.06\,\%)$ | 268,006 |
| CGS(16) | $4.60 \pm 2.56$ $(0.81 \pm 0.10\,\%)$ | $1.93 \pm 1.22$ $(0.84 \pm 0.10\,\%)$ | $1.93 \pm 1.12$ $(0.81 \pm 0.07\,\%)$ | $\mathbf{1.65 \pm 1.06}$ $(0.84 \pm 0.09\,\%)$ | $1.76 \pm 0.86$ $(0.80 \pm 0.06\,\%)$ | $1.57 \pm 0.86$ $(0.80 \pm 0.06\,\%)$ | $1.73 \pm 0.83$ $(0.80 \pm 0.05\,\%)$ | $1.45 \pm 0.77$ $(0.79 \pm 0.05\,\%)$ | 258,469 |
| CGS(32) | $\mathbf{4.39 \pm 2.67}$ $(0.85 \pm 0.09\,\%)$ | $2.00 \pm 1.18$ $(0.83 \pm 0.09\,\%)$ | $1.90 \pm 1.07$ $(0.81 \pm 0.06\,\%)$ | $2.16 \pm 1.11$ $(0.77 \pm 0.10\,\%)$ | $1.73 \pm 0.85$ $(0.81 \pm 0.05\,\%)$ | $1.24 \pm 0.48$ $(0.76 \pm 0.06\,\%)$ | $1.72 \pm 0.87$ $(0.80 \pm 0.05\,\%)$ | $1.19 \pm 0.41$ $(0.76 \pm 0.05\,\%)$ | 265,669 |
| CGS(64) | $4.55 \pm 2.60$ $(0.85 \pm 0.09\,\%)$ | $\mathbf{1.83 \pm 1.18}$ $(0.86 \pm 0.09\,\%)$ | $\mathbf{1.78 \pm 1.02}$ $(0.83 \pm 0.07\,\%)$ | $2.36 \pm 1.37$ $(0.85 \pm 0.09\,\%)$ | $\mathbf{1.68 \pm 0.82}$ $(0.83 \pm 0.05\,\%)$ | $\mathbf{1.23 \pm 0.76}$ $(0.83 \pm 0.05\,\%)$ | $\mathbf{1.59 \pm 0.78}$ $(0.82 \pm 0.05\,\%)$ | $\mathbf{1.13 \pm 0.67}$ $(0.82 \pm 0.05\,\%)$ | 280,069 |

We train three CGS models, CGS(16), CGS(32), and CGS(64), and two baseline models, SSE and IGNN, on randomly generated MDP graphs. Each MDP graph has $n_s$ nodes and each node has $n_a$ edges (i.e. the MDP has $n_s$ distinct states and $n_a$ possible actions from each state). We sample $n_s$ and $n_a$ from the discrete uniform distributions with lower and upper bounds of (20, 50) and (5, 10), respectively. We evaluate the predictive performance of the trained CGS on randomly generated GVI problems to verify the generalization capability of CGS for different $n_s$ and $n_a$. We refer to Appendix C.2 for the details of the model architectures and training schemes.

Table 1 summarizes the evaluation results of CGS and the baseline models. We report the mean-absolute-percentage error (MAPE) between the predicted state values and their true values, and the accuracy between the derived and optimal policies (in %), following Deac et al. (2020).

All the CGS models show reliable value and policy predictions for the in-training cases ($n_s = 20$) as well as for the out-of-training cases. In general, CGS with many heads shows better prediction results than the model with smaller number of heads. The two baseline models shows significant performance drops as $n_s$ and, especially, $n_a$ increase. (i.e. the number of incoming edges per node becomes larger). The results show that CGS can solve the unseen problems well (i.e., generalization). From the results, we can conclude constructing the transition map adaptively by utilizing the input graph information is a crucial factor for achieving better generalization and higher prediction accuracy of CGS, as compared to other baseline models (SSE and IGNN) that uses fixed transition maps.

**Ablation studies.** We conduct various ablation studies and confirm the following:

- **Appendix D.1**: the flexibility of $f_\theta(\cdot)$ design is beneficial to attain higher predictive performances,

- **Appendix D.2**: the input-dependency of both $\boldsymbol{A}_\theta(\mathcal{G})$ and $\boldsymbol{B}_\theta(\mathcal{G})$ is essential for generalization,

- **Appendix D.3**: the contraction factor $\gamma$ balances the predictability and computational time,

- **Appendix D.4**: the linear $\mathcal{T}_\theta(\cdot)$ has sufficient expressivity, compared to the nonlinear extension, while being robust to the hyperparameters,

- **Appendix D.5**: a relatively small of training samples ($\geq 512$) is enough to attain higher predictive performances, compared to IGNN.

- **Appendix D.6**: the iterative fixed point computation scales better than the direct method in terms of memory usage.

From the results, we confirmed the proposed design of $\mathcal{T}_\theta(\cdot)$ is effective and robust to the hyperparameters in solving GVI , and the proposed training scheme – iterative fixed point computation and

Table 2: **Graph classification results (accuracy in** %). All benchmark results are reproduced from the original papers except IGNN.

|  | IMDB-B | IMDB-M | MUTAG | PROT. | PTC | NCI1 |
|---|---|---|---|---|---|---|
| # graphs | 1000 | 1500 | 188 | 1113 | 344 | 4110 |
| # classes | 2 | 3 | 2 | 2 | 2 | 2 |
| Avg # nodes | 19.8 | 13.0 | 17.9 | 39.1 | 25.5 | 29.8 |
| PATCHY-SAN (Niepert et al., 2016) | $71.0 \pm 2.2$ | $45.2 \pm 2.8$ | $92.6 \pm 4.2$ | $75.9 \pm 2.8$ | $60.0 \pm 4.8$ | $78.6 \pm 1.9$ |
| DGCNN (Zhang et al., 2018) | 70.0 | 47.8 | 85.8 | 75.5 | 58.6 | 74.4 |
| AWL (Ivanov & Burnaev, 2018) | $74.5 \pm 5.9$ | $51.5 \pm 3.6$ | $87.9 \pm 9.8$ | – | – | – |
| GIN (Xu et al., 2018) | $75.1 \pm 5.1$ | $\mathbf{52.3 \pm 2.8}$ | $89.4 \pm 5.6$ | $76.2 \pm 2.8$ | $64.6 \pm 7.0$ | $\mathbf{82.7 \pm 1.7}$ |
| GraphNorm (Cai et al., 2020) | $\mathbf{76.0 \pm 3.7}$ | – | $\mathbf{91.6 \pm 6.5}$ | $\mathbf{77.4 \pm 4.9}$ | $\mathbf{64.9 \pm 7.5}$ | $81.4 \pm 2.4$ |
| LP-GNN (Tiezzi et al., 2020) | $71.2 \pm 4.7$ | $46.6 \pm 3.7$ | $90.5 \pm 7.0$ | $77.1 \pm 4.3$ | $64.4 \pm 5.9$ | $68.4 \pm 2.1$ |
| IGNN (ours) (Gu et al., 2020) | – | – | $78.1 \pm 11.8$ | $76.5 \pm 4.3$ | $60.8 \pm 10.3$ | $72.8 \pm 1.9$ |
| CGS(1) | $72.3 \pm 3.4$ | $49.6 \pm 3.6$ | $85.9 \pm 6.8$ | $74.1 \pm 4.1$ | $60.2 \pm 7.8$ | $75.9 \pm 1.4$ |
| CGS(4) | $73.0 \pm 1.9$ | $51.0 \pm 1.7$ | $88.4 \pm 8.0$ | $76.3 \pm 6.3$ | $64.7 \pm 6.4$ | $76.3 \pm 2.0$ |
| CGS(8) | $73.0 \pm 2.1$ | $51.1 \pm 2.2$ | $86.5 \pm 7.2$ | $76.3 \pm 4.9$ | $62.5 \pm 5.2$ | $77.6 \pm 2.2$ |
| CGS(16) | $72.8 \pm 2.5$ | $50.4 \pm 2.1$ | $88.7 \pm 6.1$ | $76.3 \pm 4.9$ | $62.9 \pm 5.2$ | $77.6 \pm 2.0$ |
| CGS(32) | $73.1 \pm 3.3$ | $50.3 \pm 1.7$ | $89.4 \pm 5.6$ | $76.0 \pm 3.2$ | $63.1 \pm 4.2$ | $77.2 \pm 2.0$ |

computing the gradient via the implicit function theorem – is practically suitable in terms of memory usage. Due to the page limit, we refer to D for the details and results of the ablation studies.

## 6.3 GRAPH CLASSIFICATION

We show that CGS can also perform general graph classification tasks accurately, where the existence or the meaning of a fixed point is hard to be clearly defined, although it is originally designed to predict quantities related to the fixed points.

We assess the graph classification performance of CGS on six graph classification benchmarks: two social-network datasets (IMDB-Binary, IMDB-Multi) and four bioinformatics datasets (MUTAG, PROTEINS, PTC, NCI1). Since the social-network datasets do not have node features, they are generated based on the node degrees following Xu et al. (2018). Also, the edge features are initialized with one vectors for all datasets. To conduct the graph classification tasks, we perform the sum readout over the outputs of CGS (i.e., summing all fixed points), and then utilize additional MLP to predict the graph labels from the readout value. We perform 10-fold cross validation and report the average and standard deviation of its accuracy for each validation fold, following the evaluation scheme of Niepert et al. (2016). We refer to Appendix C.3.1 for the details of the network architecture, training, and hyperparamter searchings.

The results in Table 2 show that the classification performance of CGS is comparable to those of other methods using fixed point iteration (LP-GNN and IGNN). Note that we reproduced the results of IGNN as the original paper used a different performance metric (IGNN (ours)). We provide the additional benchmark results in Appendix C.3.2 to compare with IGNN following their test metric. In general, CGS shows better performance than LP-GNN and IGNN, which also find the fixed points of graph convolutions on the social-network datasets where the node features are not given. From the results, CGS can be interpreted as that CGS finds "virtual fixed points" that contain the most relevant information to classify the graph labels. These results indicate that CGS has a potential as an general graph convolution layer.

## 7 CONCLUSION

We propose the convergent graph solver (CGS) as a new learning-based iterative method to compute the stationary properties of network-analytic problems. CGS generates contracting input-dependent linear iterative maps, finds the fixed points of the maps, and finally decodes the fixed points to predict the solution of the network-analytic problems. Through various network-analytic problems, we show that CGS has competitive capabilities in predicting the outputs (properties) of complex target networked systems in comparison with the other GNNs. We also show that CGS can solve general graph benchmark problems effectively, showing the potential that CGS can be used as a general graph implicit layer for processing graph structured data.

## 8 ETHIC STATEMENTS AND REPRODUCIBILITY

**Ethics statement**   We propose a deep learning method that learns iterative mappings of the target problems. As we discussed in section 1, various types of engineering, science, and societal problems are formulated in the form of fixed point-finding problems. On the bright side, we expect the proposed method to expedite scientific/engineering discoveries by serving as a fast simulation. On the other hand, as our method rooted in the idea of finding fixed points, it can be used to analyze networks and finding an adversarial or weak point of a network that can change the results of many network-analytic algorithms that are ranging from everyday usages, such as recommendation engines of commercial services, and maybe life-critical usages.

**Reproducibility**   As machine learning researchers, we consider the reproducibility of numerical results as one of the top priorities. Thus, we put a significant amount of effort into pursuing the reproducibility of our experimental results. As such, we set and tracked the random seed used for our experiments and confirmed the experiments were reproducible.

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

# Convergent Graph Solvers
## *Supplementary Material*

## Table of Contents

## A  PROOFS AND DERIVATIONS

In this section, we prove the existence of the converged hidden embedding $\boldsymbol{H}^*$ and $(I - \gamma \boldsymbol{A}_\theta)^{-1}$ and give the complement derivation of the partial derivatives that is used to train CGS.

### A.1  EXISTENCE OF CONVERGED EMBEDDING

The proposed transition map is defined as follows:

$$\mathcal{T}_\theta(\boldsymbol{H}^{[n]}; \mathcal{G}) = \gamma \boldsymbol{A}_\theta(\mathcal{G}) \boldsymbol{H}^{[n]} + \boldsymbol{B}_\theta(\mathcal{G}) \tag{A.1}$$

where $0.0 < \gamma < 1.0$, $\boldsymbol{A}_\theta(\mathcal{G}) \in \mathbb{R}^{p \times p}$ with $||\boldsymbol{A}_\theta(\mathcal{G})|| \leq 1.0$, and $\boldsymbol{B}_\theta(\mathcal{G}) \in \mathbb{R}^{p \times q}$.

We first show that the proposed transition map is contracting. Then, the existence of unique fixed point can be directly obtained by applying the Banach fixed point theorem (Banach, 1922).

**Lemma 1.** $\mathcal{T}_\theta(\boldsymbol{H}^{[n]}; \mathcal{G})$ is a $\gamma$-contraction mapping.

*Proof.* The proof is trivial. Consider the following equations:

$$
\begin{aligned}
||\mathcal{T}_\theta(\boldsymbol{H}^{[n+1]}; \mathcal{G}) - \mathcal{T}_\theta(\boldsymbol{H}^{[n]}; \mathcal{G})|| &= ||\gamma \boldsymbol{A}_\theta(\mathcal{G}) \boldsymbol{H}^{[n+1]} + \boldsymbol{B}_\theta(\mathcal{G}) - \gamma \boldsymbol{A}_\theta(\mathcal{G}) \boldsymbol{H}^{[n]} - \boldsymbol{B}_\theta(\mathcal{G})|| \\
&= \gamma ||\boldsymbol{A}_\theta(\mathcal{G})(\boldsymbol{H}^{[n+1]} - \boldsymbol{H}^{[n]})|| \\
&\leq \gamma ||\boldsymbol{A}_\theta(\mathcal{G})|| \times ||(\boldsymbol{H}^{[n+1]} - \boldsymbol{H}^{[n]})|| \\
&= \gamma ||(\boldsymbol{H}^{[n+1]} - \boldsymbol{H}^{[n]})||
\end{aligned}
$$

The inequality holds by the property of the spectral norm. Therefore, $\mathcal{T}_\theta(\boldsymbol{H}^{[n]}; \mathcal{G})$ is $\gamma$-contracting. □

### A.2  EXISTENCE OF INVERSE MATRIX

CGS can find the fixed point $H^*$ by pre-multiplying the inverse matrix of $(I - \gamma \boldsymbol{A}_\theta)$ to $B_\theta$, which is given as follows:

$$\boldsymbol{H}^* = (I - \gamma \boldsymbol{A}_\theta)^{-1} \boldsymbol{B}_\theta \tag{A.2}$$

The following lemma shows the existence of the inverse matrix.

**Lemma 2.** *With matrix $\boldsymbol{A}_\theta$, which is constructed using equation 6, and $0.0 < \gamma < 1.0$, $(I - \gamma \boldsymbol{A}_\theta)$ is invertible.*

*Proof.* According to the invertible matrix theorem, $(I - \gamma \boldsymbol{A}_\theta)$ being invertible is equivalent to $(I - \gamma \boldsymbol{A}_\theta)\boldsymbol{x} = 0$ only having the trivial solution $\boldsymbol{x} = 0$. By rewriting the equation as $\boldsymbol{x} = \gamma \boldsymbol{A}_\theta \boldsymbol{x}$ and defining the $\gamma$-contraction map $\mathcal{F}(\boldsymbol{x}) = \gamma \boldsymbol{A}_\theta \boldsymbol{x}$, we can get the following result:

$$\lim_{n \to \infty} \mathcal{F}^n(\boldsymbol{x}) = 0, \text{ for any } \boldsymbol{x}. \tag{A.3}$$

From the Banach fixed point theorem, we can conclude that the unique solution $x$ is 0. Therefore, $(I - \gamma \boldsymbol{A}_\theta)$ is invertible. □

### A.3  THE COMPLETE DERIVATION OF THE PARTIAL DERIVATIVES

The following equality shows the relationship of the gradient of the scalar-valued loss $\mathcal{L}$, the fixed point $\boldsymbol{H}^*$, and the parameters of the fixed point equation $\boldsymbol{A}_\theta(\mathcal{G})$, $\boldsymbol{B}_\theta(\mathcal{G})$:

$$\frac{\partial \mathcal{L}}{\partial (\cdot)} = \frac{\partial \mathcal{L}}{\partial \boldsymbol{H}^*} \frac{\partial \boldsymbol{H}^*}{\partial (\cdot)} \tag{A.4}$$

where $(\cdot)$ denotes $\boldsymbol{A}_\theta(\mathcal{G})$ or $\boldsymbol{B}_\theta(\mathcal{G})$. Here, $\frac{\partial \mathcal{L}}{\partial \boldsymbol{H}^*}$ are readily computable via automatic differentiation packages. However, computing $\frac{\partial \boldsymbol{H}^*}{\partial (\cdot)}$ is less straightforward since $\boldsymbol{H}^*$ and $(\cdot)$ are implicitly related via equation 6.

We reformulate the transition map (equation 6) in a root-finding form $g(\boldsymbol{H}, \boldsymbol{A}, \boldsymbol{B}) = \boldsymbol{H} - (\gamma \boldsymbol{A}\boldsymbol{H} + \boldsymbol{B})$. Here, we omit the input-dependency of $\boldsymbol{A}$ and $\boldsymbol{B}$ for the brevity of the notations. For the fixed point $\boldsymbol{H}^*$, the following equality holds:

$$\frac{\partial g(\boldsymbol{H}^*(\boldsymbol{A}, \boldsymbol{B}), \boldsymbol{A}, \boldsymbol{B})}{\partial \boldsymbol{A}} = 0 \tag{A.5}$$

We can expand the derivative by using chain rule as follows:

$$\frac{\partial g(\boldsymbol{H}^*(\boldsymbol{A}, \boldsymbol{B}), \boldsymbol{A}, \boldsymbol{B})}{\partial \boldsymbol{A}} = \frac{\partial g(\boldsymbol{H}^*, \boldsymbol{A}, \boldsymbol{B})}{\partial \boldsymbol{A}} + \frac{\partial g(\boldsymbol{H}^*, \boldsymbol{A}, \boldsymbol{B})}{\partial \boldsymbol{H}^*} \frac{\partial \boldsymbol{H}^*(\boldsymbol{A}, \boldsymbol{B})}{\boldsymbol{A}} = 0 \tag{A.6}$$

By rearranging the terms, we can get a closed-form expression of $\frac{\partial \boldsymbol{H}^*}{\partial \boldsymbol{A}}$ as follows:

$$\frac{\partial \boldsymbol{H}^*}{\partial \boldsymbol{A}} = -\left(\frac{\partial g(\boldsymbol{H}^*, \boldsymbol{A}, \boldsymbol{B})}{\partial \boldsymbol{H}^*}\right)^{-1} \frac{\partial g(\boldsymbol{H}^*, \boldsymbol{A}, \boldsymbol{B})}{\partial \boldsymbol{A}} \tag{A.7}$$

Here, we can compute $\frac{\partial g(\boldsymbol{H}^*, \boldsymbol{A}, \boldsymbol{B})}{\partial \boldsymbol{A}}$ easily by using either automatic differentiation tool or manual gradient calculation. However, in practice, directly computing $(\frac{\partial g(\boldsymbol{H}^*, \boldsymbol{A}, \boldsymbol{B})}{\partial \boldsymbol{H}^*})^{-1}$ can be problematic since it involves the inversion of Jacobian matrix. Instead, we can (1) construct a linear system whose solution is $(\frac{\partial g(\boldsymbol{H}^*, \boldsymbol{A}, \boldsymbol{B})}{\partial \boldsymbol{H}^*})^{-1}$ and solve the system via some matrix decomposition or (2) solve another fixed point iteration which converges to $(\frac{g(\boldsymbol{H}^*, \boldsymbol{A}, \boldsymbol{B})}{\partial \boldsymbol{H}^*})^{-1}$. We provide the pseudocode that shows how to compute the inverse Jacobian with the second option in section B.

The partial derivative with respect to $\boldsymbol{B}$ can be computed through the similar procedure and it is given as follows:

$$\frac{\partial \boldsymbol{H}^*}{\partial \boldsymbol{B}} = -\left(\frac{\partial g(\boldsymbol{H}^*, \boldsymbol{A}, \boldsymbol{B})}{\partial \boldsymbol{H}^*}\right)^{-1} \frac{\partial g(\boldsymbol{H}^*, \boldsymbol{A}, \boldsymbol{B})}{\partial \boldsymbol{B}} \tag{A.8}$$

## B  SOFTWARE IMPLEMENTATION

In this section, we provide a `Pytorch` style pseudocode of CGS which computes the derivatives via the backward fixed point iteration.

```python
import torch
import torch.nn as nn

class CGP(nn.Module):
    """
    Convergent Graph Propagation
    """

    def __init__(self,
                 gamma: float,
                 activation: str,
                 tol: float = 1e-6,
                 max_iter: int = 50):

        super(CGP, self).__init__()
        self.gamma = gamma
        self.tol = tol
        self.max_iter = max_iter
        self.act = getattr(nn, activation)()

        self.frd_itr = None  # forward iteration steps

    def forward(self, A, b):
        """
        :param A: A matrix [#.heads x #. edges x #. edges]
        :param b: b matrix [#.heads x #. nodes x 1]
        :return: z: Fixed points [#. heads x #. nodes]
        """

        z, self.frd_itr = self.solve_fp_eq(A, b,
                                           self.gamma,
                                           self.act,
                                           self.max_iter,
                                           self.tol)

        # re-engage autograd and add the gradient hook
        z = self.act(self.gamma * torch.bmm(A, z) + b)

        if z.requires_grad:
            y0 = self.gamma * torch.bmm(A, z) + b
            y0 = y0.detach().requires_grad_()
            z_next = self.act(y0).sum()
            z_next.backward()
            dphi = y0.grad
            J = self.gamma * (dphi * A).transpose(2, 1)

            def modify_grad(grad):
                y, bwd_itr = self.solve_fp_eq(J,
                                              grad,
                                              1.0,
                                              nn.Identity(),
                                              self.max_iter,
                                              self.tol)

                return y

            z.register_hook(modify_grad)
        z = z.squeeze(dim=-1)  # drop dummy dimension
        return z
```

```python
@staticmethod
@torch.no_grad()
def solve_fp_eq(A, b,
                gamma: float,
                act: nn.Module,
                max_itr: int,
                tol: float):
    """
    Find the fixed point of x = act(gamma * A * x + b)
    """

    x = torch.zeros_like(b, device=b.device)
    itr = 0
    while itr < max_itr:
        x_next = act(gamma * torch.bmm(A, x) + b)
        g = x - x_next
        if torch.norm(g) < tol:
            break
        x = x_next
        itr += 1
    return x, itr
```

Listing 1: CGS pseudocode

## C    DETAILS ON EXPERIMENTS

In this section, we provide the details of experiments including generation schemes and hyperparaters of models. We run all experiments on a single desktop equipped with a NVIDIA Titan X GPU and AMD Threadripper 2990WX CPU.

### C.1    PHYSICAL DIFFUSION EXPERIMENTS

#### C.1.1    DATA GENERATION

In this section, we provide the details of porous network problems.

**Diffusion Equation**    For fluid flow in porous media described by Darcy's law, equation 14 is specific to

$$\sum_{j \in \mathcal{N}(i)} \frac{\pi}{8\mu} \frac{r_{ij}^4}{l_{ij}} (p_i - p_j) = 0, \quad \forall v_i \in \mathbb{V} \setminus \partial(\mathbb{V}), \tag{A.9}$$

where $\mu$ is the dynamic viscosity of the fluid, and $r_{ij}$ and $l_{ij}$ are the radius and length of the cylindrical throat between the $i$-th and $j$-th pore chambers respectively.

**Data generation**    We generate random pore networks inside a cubic domain of width $0.1\,\mathrm{m}$ using Voronoi tessellation. We sample the pore diameters from the uniform distribution of $U(9.9 \times 10^{-3}\,\mathrm{m}, 10.1 \times 10^{-3}\,\mathrm{m})$ and assume that the fluid is water with $\mu = 10^{-3}\,\mathrm{N\,s\,m^{-2}}$ under the temperature of $298\,\mathrm{K}(25\,°\mathrm{C})$. The boundary conditions are the atmospheric pressure $(101, 325\,\mathrm{Pa})$ on the front surface of the cube, zero pressure on the back surface, and no-flux conditions on all other surfaces. We simulate the flow using `OpenPNM` (Gostick et al., 2016). We normalize the pressure values (targets) by dividing the pressure by the maximum pressures of the pores. Note that this normalization is always viable because the maximum pressure is the prescribed boundary pressure on the front surface due to the physical nature of the diffusion problem.

#### C.1.2    DETAILS OF CGS AND BASELINES

In this section, we provide the details of CGS and the baseline models. For brevity, we refer an MLP with hidden neurons $n_1$, $n_2$, ... $n_l$ for each hidden layer as MLP($n_1$, $n_2$, ..., $n_l$).

**Network architectures**

- `CGS(m)`: $f_\theta$ is a single layer attention variant of graph network (GN) (Battaglia et al., 2018) whose edge, attention, and node function are MLP(64). The output dimensions of the edge and node function are determined by the number of heads $m$. $\rho(\cdot)$ is the summation. $g_\theta$ is MLP(64, 32). All hidden activations are LeakyReLU. We set $\gamma$ as 0.5.

- `IGNN`: $f_\theta$, $g_\theta$ is the same as the one of `CGS(8)`.

- `SSE`: We modify the original SSE implementation (Dai et al., 2018) so that the model can take the edge feature as an additional input. As $g_\theta$, we use the same architecture to the one of `CGS(m)`.

- `GNN(n)`: It is the plain GNN architecture having the stacks of $n$ different GNN layers as $f_\theta$. For each GNN layer, we utilize the same GN layer architecture to the one of `CGS(m)`. $g_\theta$ is the same as the one of `CGS(m)`.

**Training details**    We train all models with the Adam optimizer (Kingma & Ba, 2014), whose learning rate is initialized as 0.001 and scheduled by the cosine annealing method (Loshchilov & Hutter, 2016). The loss function is the mean-squared error (MSE) between the model predictions and the ground truth pressures. The training graphs were generated on-fly as specified in the **data generation** paragraph. We used 32 training graphs per gradient update. On every 32 gradient update, we sample the new training graph. We train 1000 gradient steps for all models.

**Training curves**    The training curves of the CGS models and baselines are provided in figure 6.

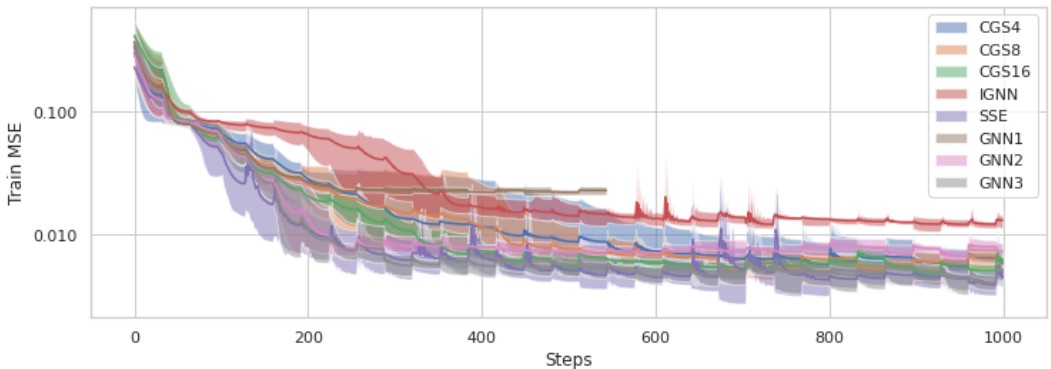

Figure 6: **Training curves of the CGS models and the baselines.** We repeat the training 5 times per each model. The solid lines show the average training MSE over the training steps. The shadow areas visualize the $\pm$ 1.0 standard deviation over the runs.

## C.2 GRAPH VALUE ITERATION EXPERIMENTS

In this section, we provide the details of data generation, the CGS models and baseline architecture and their training schemes.

### C.2.1 DETAILS OF GVI DATA GENERATION

**Data generation** We generate the MDP graph by randomly sampling $n_a$ out-warding edges for all nodes. The discount factor of MDP $\alpha$ is 0.9. Rewards are sampled from the uniform distribution whose upper and lower bounds are 1.0 and -1.0 respectively. The true state-values (labels) are computed by iteratively employing the exact analytical Bellman operator (equation 5) until the state-values converge (i.e., value iteration). The convergence tolerance is 0.001.

### C.2.2 DETAILS OF CGS AND BASELINES

**Network architectures**

- `CGS(m)`: $f_\theta$ is a three layer attention GN network as whose edge, attention, and node function are MLP(128). The output dimensions of the edge and node function are determined by the number of heads $m$. $\rho(\cdot)$ is the summation. $g_\theta$ is MLP(64, 32). All hidden activations are LeakyReLU. We set $\gamma$ as 0.5.
- `IGNN`: $f_\theta$, $g_\theta$ is the same as the one of `CGS(32)`.
- `SSE`: We modify the original SSE implementation (Dai et al., 2018) so that the model can take the edge feature as an additional input. As $g_\theta$, we use the same architecture to the one of `CGS(m)`.

**Training details** We train all models with the Adam optimizer whose learning rate is initialized as 0.001 and scheduled by the cosine annealing method. The loss function is MSE between the model predictions and the ground truth state-values. The training graphs are generated on-fly as specified in the **data generation** paragraph. We use 64 training graphs per gradient update. On every 32 gradient graphs, we sample the new training graph. We train 5000 gradient steps for all models.

## C.3 GRAPH CLASSIFICATION EXPERIMENTS

### C.3.1 EXPERIMENT DETAILS AND HYPERPARAMETERS

In this section, we explain the network architecture and training details of the six graph benchmark problems. Across all the benchmark dataset, we use the dataset implementation of `DGL` (Wang et al., 2019) and cross-validation indices generated with `Scipy` (Virtanen et al., 2020). We set the contraction factor $\gamma$ as 0.5. We train all models with the Adam optimizer whose learning rate is initialized as 0.001 and scheduled by the cosine annealing method for 500 (100 for `NCI1` dataset

Table 3: **Graph classification results (accuracy in %).**

|  | IMDB-B | IMDB-M | MUTAG | PROT. | PTC | NCI1 |
|---|---|---|---|---|---|---|
| # graphs | 1000 | 1500 | 188 | 1113 | 344 | 4110 |
| # classes | 2 | 3 | 2 | 2 | 2 | 2 |
| Avg # nodes | 19.8 | 13.0 | 17.9 | 39.1 | 25.5 | 29.8 |
| IGNN (Gu et al., 2020) | − | − | $89.3 \pm 6.7$ | $77.7 \pm 3.4$ | $70.1 \pm 5.6$ | $\mathbf{80.5 \pm 1.9}$ |
| CGS(4) | $76.0 \pm 2.0$ | $62.3 \pm 0.5$ | $93.5 \pm 7.7$ | $\mathbf{81.3 \pm 4.5}$ | $\mathbf{75.3 \pm 6.3}$ | $77.86 \pm 1.8$ |
| CGS(8) | $76.0 \pm 2.3$ | $62.7 \pm 0.5$ | $94.2 \pm 5.2$ | $81.0 \pm 4.4$ | $74.0 \pm 5.5$ | $78.9 \pm 2.0$ |
| CGS(16) | $\mathbf{76.4 \pm 2.0}$ | $\mathbf{63.4 \pm 0.4}$ | $94.6 \pm 4.3$ | $80.2 \pm 4.4$ | $75.0 \pm 7.0$ | $79.1 \pm 1.6$ |
| CGS(32) | $76.4 \pm 3.0$ | $59.4 \pm 1.0$ | $\mathbf{95.7 \pm 4.2}$ | $80.1 \pm 4.4$ | $75.0 \pm 6.0$ | $79.1 \pm 2.1$ |

due to the large datset size) epochs with 128 mini-batch size. We set the random seed of `Scipy`, `Pytorch` Paszke et al. (2019), and `DGL` as 0.

**Hyperparameter tuning**    Due to our limited computational resources, we search at most 10 different pairs of hyperparameters for each dataset. To find the initial hyperparemters, we first tune `CGS(4)` on MUTAG, which is the smallest dataset. For each benchmark dataset, we start hyperparmeter tunings from the hyperparameters that are used for MUTAG-`CGS(4)` and tune the activation functions of $f_\theta$ and $g_\theta$, the number of GN layers in $f_\theta$, the number of layers of $g_\theta$, and the dropout rate of $g_\theta$.

**Network architectures**

- **IMDB-Binary**: $f_\theta$ is a two layer attention GN network as whose edge, attention, and node function are MLP(128). The output dimensions of the edge and node function are determined by the number of heads $m$. The hidden dimensions of edge, and node are 64. $\rho(\cdot)$ is the summation. $g_\theta$ is MLP(64, 32). All hidden activations are Swish Ramachandran et al. (2017).

- **IMBD-Multi**: The same as **IMDB-Binary**.

- **MUTAG**: The same as **IMDB-Binary**. All hidden activations are LeakyReLU.

- **PROTEINS**: The same as the **MUTAG**. Apply dropout Srivastava et al. (2014) with probability 0.2 after the activation functions of $g_\theta$.

- **NCI1**: The same as **IMDB-Binary**.

### C.3.2    EXTENDED BENCHMARK RESULTS

We provide the benchmark results of `CGS` with the test metric that is used from (Gu et al., 2020). The test metric is to calculate the average maximal test accuracies over the test folds. `CGS` shows better predictive accuracies compared to `IGNN`.

Table 4: **Graph Value Iteration results over the different $f_\theta(\cdot)$ architecture.** We report the average MAPE and policy prediction accuracies (in %) of different $n_s$ and $n_a$ combinations. All metrics are measured per graph. $\pm$ shows the standard deviation of the metrics.

| $n_s$ | 20 | | 50 | | 75 | | 100 | | #. params |
|---|---|---|---|---|---|---|---|---|---|
| $n_a$ | 5 | 10 | 10 | 15 | 10 | 15 | 10 | 15 | |
| 1-CGS(16) | $7.33 \pm 4.59$ $(0.77 \pm 0.11 \%)$ | $3.38 \pm 2.00$ $(0.75 \pm 0.11 \%)$ | $3.21 \pm 1.70$ $(0.72 \pm 0.08 \%)$ | $3.62 \pm 2.08$ $(0.73 \pm 0.11 \%)$ | $3.23 \pm 1.60$ $(0.72 \pm 0.06 \%)$ | $2.60 \pm 1.21$ $(0.70 \pm 0.06 \%)$ | $3.15 \pm 1.47$ $(0.71 \pm 0.06 \%)$ | $2.66 \pm 1.16$ $(0.71 \pm 0.06 \%)$ | 10,786 |
| 2-CGS(16) | $6.86 \pm 4.46$ $(0.81 \pm 0.10 \%)$ | $2.78 \pm 1.75$ $(0.81 \pm 0.10 \%)$ | $2.17 \pm 1.14$ $(0.79 \pm 0.07 \%)$ | $2.48 \pm 1.54$ $(0.78 \pm 0.10 \%)$ | $1.99 \pm 0.92$ $(0.78 \pm 0.06 \%)$ | $1.34 \pm 0.61$ $(0.76 \pm 0.06 \%)$ | $1.89 \pm 0.72$ $(0.77 \pm 0.06 \%)$ | $1.28 \pm 0.58$ $(0.77 \pm 0.05 \%)$ | 93,347 |
| 3-CGS(16) | $6.12 \pm 3.97$ $(0.81 \pm 0.10 \%)$ | $2.65 \pm 1.68$ $(0.83 \pm 0.09 \%)$ | $2.33 \pm 1.36$ $(0.80 \pm 0.07 \%)$ | $3.14 \pm 1.79$ $(0.84 \pm 0.09 \%)$ | $2.23 \pm 1.18$ $(0.80 \pm 0.06 \%)$ | $2.55 \pm 1.16$ $(0.80 \pm 0.06 \%)$ | $2.13 \pm 1.10$ $(0.79 \pm 0.05 \%)$ | $2.63 \pm 1.12$ $(0.80 \pm 0.05 \%)$ | 175,908 |
| 4-CGS(16) | $4.42 \pm 3.07$ $(0.88 \pm 0.08 \%)$ | $1.92 \pm 1.20$ $(0.86 \pm 0.09 \%)$ | $1.72 \pm 1.02$ $(0.84 \pm 0.06 \%)$ | $2.09 \pm 1.28$ $(0.84 \pm 0.09 \%)$ | $1.60 \pm 0.91$ $(0.84 \pm 0.05 \%)$ | $1.25 \pm 0.77$ $(0.83 \pm 0.05 \%)$ | $1.57 \pm 0.80$ $(0.82 \pm 0.05 \%)$ | $1.19 \pm 0.72$ $(0.83 \pm 0.05 \%)$ | 258,469 |

## D ABLATION STUDIES

In this section, we provide the results of the ablation studies in GVI. The ablation studies were done to understand

- the effect of $f_\theta(\cdot)$ architecture,
- the effect of transition map design,
- the effect of contraction factor $\gamma$,
- the effect of employing the non-linear transition maps,
- the sample efficiency of CGS.

We also provide the runtime comparisons of direct inversion and iterative methods to find the fixed points of hidden embedding.

### D.1 EFFECT OF $f_\theta(\cdot)$ ARCHITECTURE

As the $A_\theta(\mathcal{G})$ and $B_\theta(\mathcal{G})$ generation schemes of CGS allow the employment of the arbitrary architecture of GNN as $f_\theta(\cdot)$, CGS has different predictive performances depending on the architectural selection of $f_\theta(\cdot)$.

Here, we investigate the effect of the number of GN layers in $f_\theta(\cdot)$ to the predictive performance of CGS for the GVI problems. The number of the independent GN layers controls the range of information when CGS constructs the transition maps. That is, a larger number of GN layers allows the information to be gathered from far neighborhoods while constructing $A_\theta(\mathcal{G})$ and $B_\theta(\mathcal{G})$.

Table 4 shows the predictive performances of CGS models with the different number of GN layers in $f_\theta(\cdot)$. The model with $n$ GN layers is referred to as $n$-CGS(16). In general, the model with a larger number of GN layers performs better. These results highlight that allowing the flexibility in $f_\theta(\cdot)$ can be practically beneficial when we derive the model with guaranteed convergence.

### D.2 DESIGN OF THE INPUT-DEPENDENT TRANSITION MAPS

The transition map $\mathcal{T}_\theta(\cdot)$ of CGS is given as follows:

$$\mathcal{T}_\theta(\boldsymbol{H}^{[n]}; \mathcal{G}) = \gamma \boldsymbol{A}_\theta(\mathcal{G})\boldsymbol{H}^{[n]} + \boldsymbol{B}_\theta(\mathcal{G}) \tag{A.10}$$

By restricting $A_\theta(\mathcal{G})$ and $B_\theta(\mathcal{G})$ to be input independent, we can consider simpler variants of CGS. We can consider three variants which has (1) input-independent $A$ and $B$; (2) input-independent $A$ and input-dependent $B$; (3) input-dependent $A$ and input-independent $B$. Out of three variants, (1) cannot differentiate different $\mathcal{G}$ and (2) is similar to Gu et al. (2020). Therefore, we provide the GVI results of (3) - Fixed $B$ CGS - in here. From Table 5, we can confirm that having input-dependent $A$ and $B$ in $\mathcal{T}_\theta(\cdot)$ shows best performances.

### D.3 EFFECT OF CONTRACTION PARAMETER $\gamma$

The contraction parameter $\gamma$, as a hyperparameter, changes the fixed point (because it changes the linear map) and controls the rate of convergent speed as well. We investigate the effect of $\gamma$ to the

Table 5: **Transition map ablation.** We report the average MAPE and policy prediction accuracies (in %) of different $n_s$ and $n_a$ combinations with 500 repeats per combination. All metrics are measured per graph. $\pm$ shows the standard deviation of the metrics.

| $n_s$ | 20 | | 50 | | 75 | | 100 | | #. params |
|---|---|---|---|---|---|---|---|---|---|
| $n_a$ | 5 | 10 | 10 | 15 | 10 | 15 | 10 | 15 | |
| IGNN | $13.87 \pm 4.69$ $(0.68 \pm 0.12\%)$ | $28.38 \pm 1.77$ $(0.63 \pm 0.13\%)$ | $28.13 \pm 1.40$ $(0.61 \pm 0.08\%)$ | $29.44 \pm 1.35$ $(0.62 \pm 0.13\%)$ | $28.21 \pm 1.29$ $(0.60 \pm 0.07\%)$ | $29.20 \pm 0.88$ $(0.60 \pm 0.07\%)$ | $28.00 \pm 1.15$ $(0.60 \pm 0.06\%)$ | $29.17 \pm 0.81$ $(0.60 \pm 0.06\%)$ | 268,006 |
| Fixed $B$ CGS(16) | $10.38 \pm 6.87$ $(0.70 \pm 0.12\%)$ | $5.30 \pm 2.97$ $(0.67 \pm 0.12\%)$ | $6.60 \pm 2.71$ $(0.66 \pm 0.08\%)$ | $7.81 \pm 3.01$ $(0.68 \pm 0.12\%)$ | $6.74 \pm 2.58$ $(0.65 \pm 0.06\%)$ | $10.61 \pm 1.78$ $(0.64 \pm 0.07\%)$ | $6.87 \pm 2.21$ $(0.66 \pm 0.05\%)$ | $10.98 \pm 1.66$ $(0.65 \pm 0.06\%)$ | 258,485 |
| Fixed $B$ CGS(32) | $9.52 \pm 6.45$ $(0.70 \pm 0.12\%)$ | $6.12 \pm 3.33$ $(0.66 \pm 0.13\%)$ | $8.38 \pm 2.96$ $(0.65 \pm 0.08\%)$ | $9.00 \pm 3.17$ $(0.67 \pm 0.12\%)$ | $8.73 \pm 2.79$ $(0.65 \pm 0.06\%)$ | $12.59 \pm 1.82$ $(0.63 \pm 0.07\%)$ | $8.99 \pm 2.36$ $(0.65 \pm 0.06\%)$ | $13.10 \pm 1.70$ $(0.63 \pm 0.06\%)$ | 265,701 |
| Fixed $B$ CGS(64) | $9.68 \pm 6.56$ $(0.70 \pm 0.12\%)$ | $6.25 \pm 3.42$ $(0.68 \pm 0.12\%)$ | $7.84 \pm 2.90$ $(0.66 \pm 0.08\%)$ | $9.74 \pm 3.26$ $(0.69 \pm 0.12\%)$ | $8.16 \pm 2.74$ $(0.64 \pm 0.06\%)$ | $11.91 \pm 1.80$ $(0.64 \pm 0.07\%)$ | $8.46 \pm 2.33$ $(0.64 \pm 0.06\%)$ | $12.38 \pm 1.69$ $(0.63 \pm 0.06\%)$ | 280,133 |
| CGS(16) | $4.60 \pm 2.56$ $(0.81 \pm 0.10\%)$ | $1.93 \pm 1.22$ $(0.84 \pm 0.10\%)$ | $1.93 \pm 1.12$ $(0.81 \pm 0.07\%)$ | $\mathbf{1.65 \pm 1.06}$ $(0.84 \pm 0.09\%)$ | $1.76 \pm 0.86$ $(0.80 \pm 0.06\%)$ | $1.57 \pm 0.86$ $(0.80 \pm 0.06\%)$ | $1.73 \pm 0.83$ $(0.80 \pm 0.05\%)$ | $1.45 \pm 0.77$ $(0.79 \pm 0.05\%)$ | 258,469 |
| CGS(32) | $\mathbf{4.39 \pm 2.67}$ $(0.85 \pm 0.09\%)$ | $2.00 \pm 1.18$ $(0.83 \pm 0.09\%)$ | $1.90 \pm 1.07$ $(0.81 \pm 0.06\%)$ | $2.16 \pm 1.11$ $(0.77 \pm 0.10\%)$ | $1.73 \pm 0.85$ $(0.81 \pm 0.05\%)$ | $1.24 \pm 0.48$ $(0.76 \pm 0.06\%)$ | $1.72 \pm 0.87$ $(0.80 \pm 0.05\%)$ | $1.19 \pm 0.41$ $(0.76 \pm 0.05\%)$ | 265,669 |
| CGS(64) | $4.55 \pm 2.60$ $(0.85 \pm 0.09\%)$ | $\mathbf{1.83 \pm 1.18}$ $(0.86 \pm 0.09\%)$ | $\mathbf{1.78 \pm 1.02}$ $(0.83 \pm 0.07\%)$ | $2.36 \pm 1.37$ $(0.85 \pm 0.09\%)$ | $\mathbf{1.68 \pm 0.82}$ $(0.83 \pm 0.05\%)$ | $\mathbf{1.23 \pm 0.76}$ $(0.83 \pm 0.05\%)$ | $\mathbf{1.59 \pm 0.78}$ $(0.82 \pm 0.05\%)$ | $\mathbf{1.13 \pm 0.67}$ $(0.82 \pm 0.05\%)$ | 280,069 |

Table 6: $\gamma$ **ablations.** We report the average MAPE and policy prediction accuracies (in %) of different $n_s$ and $n_a$ combinations with 500 repeats per combination. All metrics are measured per graph. $\pm$ shows the standard deviation of the metrics. All metrics are measured per graph.

| $n_s$ | 20 | | 50 | | 75 | | 100 | | #. params |
|---|---|---|---|---|---|---|---|---|---|
| $n_a$ | 5 | 10 | 10 | 15 | 10 | 15 | 10 | 15 | |
| $\gamma = 0.3$ | $5.93 \pm 3.54$ $(0.83 \pm 0.09\%)$ | $2.50 \pm 1.41$ $(0.83 \pm 0.09\%)$ | $2.31 \pm 1.17$ $(0.82 \pm 0.06\%)$ | $1.63 \pm 1.01$ $(0.82 \pm 0.09\%)$ | $2.43 \pm 1.11$ $(0.81 \pm 0.06\%)$ | $1.44 \pm 0.66$ $(0.80 \pm 0.05\%)$ | $2.45 \pm 1.03$ $(0.82 \pm 0.05\%)$ | $1.52 \pm 0.64$ $(0.80 \pm 0.05\%)$ | 280,069 |
| $\gamma = 0.5$ | $4.55 \pm 2.60$ $(0.85 \pm 0.09\%)$ | $1.83 \pm 1.18$ $(0.86 \pm 0.09\%)$ | $1.78 \pm 1.02$ $(0.83 \pm 0.07\%)$ | $2.36 \pm 1.37$ $(0.85 \pm 0.09\%)$ | $1.68 \pm 0.82$ $(0.83 \pm 0.05\%)$ | $1.23 \pm 0.76$ $(0.83 \pm 0.05\%)$ | $1.59 \pm 0.78$ $(0.82 \pm 0.05\%)$ | $1.13 \pm 0.67$ $(0.82 \pm 0.05\%)$ | 280,069 |
| $\gamma = 0.7$ | $3.68 \pm 2.60$ $(0.88 \pm 0.08\%)$ | $1.68 \pm 1.08$ $(0.86 \pm 0.09\%)$ | $1.60 \pm 0.89$ $(0.85 \pm 0.06\%)$ | $1.49 \pm 0.97$ $(0.84 \pm 0.08\%)$ | $1.48 \pm 0.75$ $(0.84 \pm 0.05\%)$ | $1.17 \pm 0.54$ $(0.83 \pm 0.05\%)$ | $1.46 \pm 0.66$ $(0.84 \pm 0.04\%)$ | $1.19 \pm 0.49$ $(0.82 \pm 0.05\%)$ | 280,069 |

predictive performance of CGS. We test three $\gamma = 0.3, 0.5, 0.7$ and the experimental results are given in Table 6. The CGS model with smaller $\gamma$ tends to have higher prediction errors compared to the larger $\gamma$. However, using larger $\gamma$ enlargers the number of iterative steps. In this regard, we use $\gamma = 0.5$ to balance the computational speed and the predictive performance of CGS.

### D.4 COMPARISONS TO THE NON-LINEAR ITERATIVE MAPS

A natural question to the linear iterative map of CGS is "can we achieve a performance gain if we employ non-linear contracting iterative maps?" To answer the question, we provide the extended experiment results.

The analysis of the existence and uniqueness of fixed points still holds when CGS employs component-wise non-expansive (CONE) activation (e.g., ReLU, LeakyReLU, Tanh, Swish, Mish) to the outputs of $\mathcal{T}_\theta(\cdot)$ given as follows:

$$\mathcal{T}_\theta(\boldsymbol{H}^{[n]}; \mathcal{G}) = \phi(\gamma \boldsymbol{A}_\theta(\mathcal{G})\boldsymbol{H}^{[n]} + \boldsymbol{B}_\theta(\mathcal{G})) \tag{A.11}$$

where $\phi(\cdot)$ is a CONE activation. The gradient of loss w.r.t $\boldsymbol{A}_\theta(\mathcal{G})$ and $\boldsymbol{B}_\theta(\mathcal{G})$ can be computed similarly to the non-linear activation cases as described in Appendix B.

We compare the GVI results of linear CGS to the non-linear CGS utilizing LeakyReLU as $\phi(\cdot)$. Table 7 shows the GVI experiment results. The linear CGS, CGS($m$), and non-linear CGS, nl-CGS($m$), shows similar predictive performance on GVI experiments in general. From these experiments, we can observe that the non-linear extension of CGS does not give significant performance gain.

Furthermore, the training of non-linear CGSs can be challenging as (1) they exhibit higher variance in loss while training, and (2) the choice of non-linearity can severely change the performance of the entire model. For instance, the rectifying units such as ReLU and LeakyReLU can result in the premature termination of the iterative schemes of CGS when $\boldsymbol{B}_\theta(\mathcal{G})$ has large negative values. Bounded activation such as Tanh limits the range of hidden embeddings to a certain range.

### D.5 SAMPLE EFFICIENCY OF CGS

We assumed the training graph and corresponding labels are easily sampled while we solving physical diffusion problem and GVI. For some practical cases, this assumptions cannot be made. Hence, we investigate the sample efficiency of CGS and our closest baseline IGNN(Gu et al., 2020).

Table 7: **Graph Value Iteration results over the non-linear and linear transition maps.** We report the average MAPE and policy prediction accuracies (in $\%$) of different $n_s$ and $n_a$ combinations. All metrics are measured per graph. $\pm$ shows the standard deviation of the metrics.

| $n_s$ | 20 | | 50 | | 75 | | 100 | | #. params |
|---|---|---|---|---|---|---|---|---|---|
| $n_a$ | 5 | 10 | 10 | 15 | 10 | 15 | 10 | 15 | |
| nl-CGS(8) | $4.73 \pm 2.67$ $(0.80 \pm 0.10\,\%)$ | $3.47 \pm 1.96$ $(0.84 \pm 0.10\,\%)$ | $3.57 \pm 1.87$ $(0.81 \pm 0.07\,\%)$ | $4.92 \pm 1.64$ $(0.83 \pm 0.09\,\%)$ | $3.38 \pm 1.54$ $(0.81 \pm 0.06\,\%)$ | $4.58 \pm 1.12$ $(0.82 \pm 0.05\,\%)$ | $3.21 \pm 1.38$ $(0.81 \pm 0.05\,\%)$ | $4.61 \pm 1.04$ $(0.82 \pm 0.05\,\%)$ | 254,869 |
| nl-CGS(16) | $4.56 \pm 2.97$ $(0.83 \pm 0.09\,\%)$ | $2.03 \pm 1.25$ $(0.85 \pm 0.09\,\%)$ | $1.93 \pm 0.97$ $(0.82 \pm 0.06\,\%)$ | $1.40 \pm 0.89$ $(0.85 \pm 0.09\,\%)$ | $1.72 \pm 0.78$ $(0.82 \pm 0.05\,\%)$ | $1.17 \pm 0.52$ $(0.81 \pm 0.05\,\%)$ | $1.67 \pm 0.63$ $(0.81 \pm 0.05\,\%)$ | $1.19 \pm 0.48$ $(0.80 \pm 0.05\,\%)$ | 258,469 |
| nl-CGS(32) | $4.55 \pm 2.72$ $(0.85 \pm 0.10\,\%)$ | $3.75 \pm 1.82$ $(0.83 \pm 0.09\,\%)$ | $3.88 \pm 1.77$ $(0.79 \pm 0.07\,\%)$ | $4.58 \pm 1.50$ $(0.80 \pm 0.11\,\%)$ | $3.82 \pm 1.44$ $(0.78 \pm 0.06\,\%)$ | $4.33 \pm 1.02$ $(0.76 \pm 0.06\,\%)$ | $3.76 \pm 1.32$ $(0.78 \pm 0.05\,\%)$ | $4.34 \pm 0.97$ $(0.76 \pm 0.05\,\%)$ | 265,669 |
| nl-CGS(64) | $4.99 \pm 3.20$ $(0.82 \pm 0.10\,\%)$ | $2.00 \pm 1.21$ $(0.84 \pm 0.09\,\%)$ | $1.93 \pm 0.97$ $(0.81 \pm 0.06\,\%)$ | $1.60 \pm 1.05$ $(0.84 \pm 0.09\,\%)$ | $1.68 \pm 0.73$ $(0.81 \pm 0.05\,\%)$ | $1.18 \pm 0.66$ $(0.81 \pm 0.05\,\%)$ | $1.62 \pm 0.63$ $(0.81 \pm 0.05\,\%)$ | $1.13 \pm 0.59$ $(0.81 \pm 0.05\,\%)$ | 280,069 |
| CGS(8) | $3.81 \pm 2.76$ $(0.88 \pm 0.09\,\%)$ | $2.42 \pm 1.51$ $(0.88 \pm 0.08\,\%)$ | $2.23 \pm 1.40$ $(0.85 \pm 0.06\,\%)$ | $3.03 \pm 1.39$ $(0.84 \pm 0.09\,\%)$ | $2.03 \pm 1.20$ $(0.84 \pm 0.05\,\%)$ | $2.00 \pm 0.91$ $(0.80 \pm 0.05\,\%)$ | $1.93 \pm 1.09$ $(0.84 \pm 0.05\,\%)$ | $1.90 \pm 0.85$ $(0.80 \pm 0.05\,\%)$ | 254,869 |
| CGS(16) | $4.24 \pm 2.51$ $(0.84 \pm 0.10\,\%)$ | $3.16 \pm 1.81$ $(0.86 \pm 0.09\,\%)$ | $2.92 \pm 1.67$ $(0.83 \pm 0.06\,\%)$ | $4.26 \pm 1.44$ $(0.83 \pm 0.10\,\%)$ | $2.70 \pm 1.38$ $(0.83 \pm 0.05\,\%)$ | $3.06 \pm 1.01$ $(0.82 \pm 0.05\,\%)$ | $2.60 \pm 1.26$ $(0.83 \pm 0.05\,\%)$ | $2.95 \pm 0.95$ $(0.82 \pm 0.05\,\%)$ | 258,469 |
| CGS(32) | $4.34 \pm 2.83$ $(0.85 \pm 0.09\,\%)$ | $2.10 \pm 1.26$ $(0.83 \pm 0.09\,\%)$ | $1.95 \pm 1.04$ $(0.81 \pm 0.06\,\%)$ | $2.15 \pm 1.18$ $(0.78 \pm 0.11\,\%)$ | $1.69 \pm 0.80$ $(0.80 \pm 0.05\,\%)$ | $1.20 \pm 0.47$ $(0.76 \pm 0.06\,\%)$ | $1.61 \pm 0.70$ $(0.80 \pm 0.05\,\%)$ | $1.17 \pm 0.40$ $(0.76 \pm 0.05\,\%)$ | 265,669 |
| CGS(64) | $4.59 \pm 2.82$ $(0.84 \pm 0.09\,\%)$ | $1.93 \pm 1.25$ $(0.85 \pm 0.09\,\%)$ | $1.85 \pm 0.99$ $(0.83 \pm 0.06\,\%)$ | $2.38 \pm 1.38$ $(0.85 \pm 0.09\,\%)$ | $1.60 \pm 0.76$ $(0.83 \pm 0.06\,\%)$ | $1.17 \pm 0.75$ $(0.83 \pm 0.05\,\%)$ | $1.53 \pm 0.64$ $(0.82 \pm 0.05\,\%)$ | $1.12 \pm 0.66$ $(0.82 \pm 0.05\,\%)$ | 280,069 |

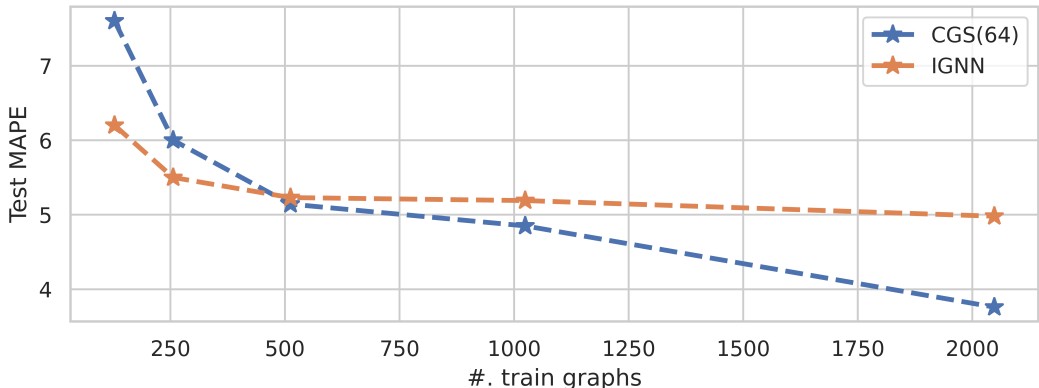

Figure 7: **Sample efficiency**

We prepare 2048 training and 2048 test graphs and their labels. We then train `CGS` and `IGNN` by using the first 128, 256, 512, 1024, and 2048 training graphs. As shown in figure 7, we can confirm that the trade-off between the sample efficiency and expressivities. `IGNN` only utilizes the input-dependent bias terms in the transition maps; thus, such structural assumptions can serve as an effective regularizer when the training samples are limited. However, `IGNN` shows more minor improvements along with the number of training samples. On the other hand, `CGS` performs worse than `IGNN` when the training graphs are limited, but it starts to outperform `IGNN` as more training samples are used. Finally, when the number of training samples increases ($\geq 512$ graphs), `CGS` outperforms `IGNN` significantly.

## D.6    RUNTIME COMPARISONS OF DIRECT INVERSION AND ITERATIVE METHODS

In this paragraph, we provide the experimental results that shows the runtime of `CGS(2)` models which solve the fixed point equation via direct inversion and iterative methods. For all size of GVI graphs, we test the models 100 times with $n_a = 5$. As shown in figure 8, for small graphs $n_s \leq 4000$, solving the fixed point equation with the direct inversion is faster than solving it with the iterative scheme. However, the direct inversion scales worse than iterative method.

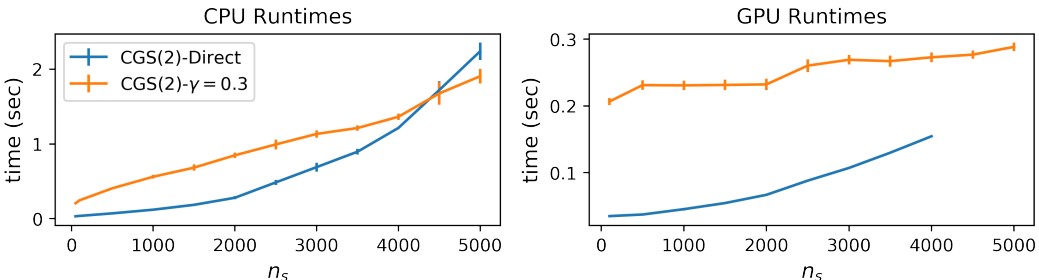

Figure 8: **Runtimes of CGS models.** For GPU experiments, the memory usage of the direct method with $n_s \geq 4000$ exceeds 24GB VRAM.

