# OpenReview forum: "Convergent Graph Solvers"
_ICLR.cc/2022/Conference — ICLR 2022 Poster_

### Official Review · Reviewer_A3tH · 2021-10-20

**Correctness:** 4
**Technical Novelty And Significance:** 3
**Empirical Novelty And Significance:** 3
**Recommendation:** 8
**Confidence:** 4

**Main Review:**

The proposed approach is technically sound and empirically evaluated on a range of experiments. The paper is generally well-written, easy to follow, and contains enough technical details to understand the underlying principles. I found that I had to look at the appendix a couple of times for important details. This includes for example the experimental details or the discussion on the runtime comparison of direct inversion and iterative methods. I think that it would have been beneficial to briefly summarize those details in the main paper.

The paper proposes a three-step procedure that is guaranteed to converge by construction. This approach is well motivated and seems promising. At first, one might wonder whether the constructed maps are expressive enough. Various experiments suggest that this is often the case. The ablation studies in the appendix corroborate the claim regarding computational complexity and suggest that the proposed method performs particularly well for problems with a large number of samples.

**Summary Of The Paper:**

This paper proposes a graph neural network termed "Convergent Graph Solvers" that computes the stationary distribution of stochastic matrices. This is achieved by an implicit definition of a linear system and by computing the fixed point of that system. The paper shows on a range of problems that the proposed approach works well and that it is particularly fast for smaller graphs.

**Summary Of The Review:**

The paper is well written, clearly explains the underlying assumptions and limitations, and shows promising results on a range of experiments.

---

> ### Author Response · Authors · 2021-11-17
> **Reply for the review A3tH**
>
> Thank you for your sincere review for our manuscript.
>
> We will update the current manuscript so that the new version will contain the summary of the details of experiments and ablation study results.

---

> > ### Comment · Reviewer_A3tH · 2021-11-26
> > **Thanks for your rebuttal**
> >
> > Thanks for modifying the manuscript accordingly. I will keep my score as accept.

---

### Official Review · Reviewer_R2aV · 2021-10-29

**Correctness:** 4
**Technical Novelty And Significance:** 4
**Empirical Novelty And Significance:** Not applicable
**Recommendation:** 8
**Confidence:** 4

**Main Review:**

**Strong points:**

The paper is well-written and motivates the choice of the linear transition map from theoretical prospective. The fact that the transition map should depend on the input graph is Interesting and has not been explored by the previous works.

The paper also provides extensive ablation studies on their modelling choices as well as the snippet of the code for the model.

**Weak points:**

It is a bit odd to compare the CGS across the number of heads, while GNN results are provided for a different number of message-passing layers. This particularly makes the figure 3 to be misleading. CGS and GNN will have a similar number of parameters if they have the same number of layers and output heads, if I understand correctly. I see that CGS results for different layers are provided in the supplement, but it will be handy to put them into the same table and compare CGS and GNN across the same parameters.

The similar issue is with SSE and IGNN models, especially since the architecture is slightly different across experiments. It would be handy to mark which CGS(m) is the most direct comparison to SSE and IGNN in terms of number of layers and heads.

**Additional questions:**

It is intriguing that implicit GNN models can handle long-range interactions without increasing the depth of the GNN. Do you have a hypothesis why?

It is odd that more output heads result in better performance. The Implicit Layers tutorial by Zico Kolter (http://implicit-layers-tutorial.org/deep_equilibrium_models/, section “One (implicit) layer is all you need”) demonstrates that a single equilibrium layer is equivalent to modelling a “stack” of several equilibrium layers (or multiple GNN heads). Can it be that the increased performance is simply due to higher output dimensionality?

Minor: I would be interested to see the runtime comparison between the GNN models and CGS, as CGS requires up to 50 iterations to converge (according to the code snippet).



**Summary Of The Paper:**

The paper introduces the equilibrium GNN-based model with a linear transition map. The transition map is made contracting to ensure that the fixed point exists and is unique. The paper provides the extensive comparisons and ablation studies for the proposed model.


**Summary Of The Review:**

The paper uses fixed-point iteration to find the equilibrium point of the graph network and constructs the transition map to be contracting and input-dependent to ensure a unique fixed point. The paper provides extensive comparisons with other equilibrium models and on different types of graphs. It is a clear accept for me.

---

> ### Author Response · Authors · 2021-11-17
> **Reply for the review R2aV**
>
> Thank you for your sincere inputs to us. We hereby provide the clarifications for your questions.
>
> **About the model comparisons**
>
> Before we start the explanations, we give a brief summary of the network architectures of CGS and the baseline models. To give a clearer explanation, we consider the network as a serial connection of encoder, iterative module, and decoder.
>
> - CGS uses 1 Layer attention GN as the encoder, the iterative module (fixed points layer) constructed from the output of the encoder, and a multi-layer perceptron (MLP) as the decoder. Note that we use one encoder for generating the parameters of multiple transition maps (heads). Hence, introducing more heads does not increase the number of GNN layers but only increases the output dimensions of the encoder.
> - IGNN uses 1 Layer attention GN as the encoder, and one layer IGNN [cite] layer as the iterative module, and a multi-layer perceptron (MLP) as the decoder.
> - SSE uses 1 Layer attention GN as the iterative module and a multi-layer perceptron (MLP) as the decoder. This is the design the original paper suggested.
> - GNN($n$) uses $n$ layered attention GN as the encoder and a multi-layer perceptron (MLP) as the decoder.
>
> To summarize, CGS and SSE use a single GNN layer in the encoder; IGNN uses two GNN layers, one for the encoder and another for decoder; GNN($n$) uses $n$ GNN layers. We agree with your suggestion that Figure (3) can be misleading. We will revise the figure so that the x-axis denotes the number of GNN layers rather than the number of entire parameters of the networks.
>
> **Hypothesis on long-range interactions enabled by CGS without increasing the depth of the GNN**
>
> We assume that the characteristic of a target system emerges from its equilibrium state (i.e., steady-state), often realized by the long-range of the interaction of the system component. CGS is designed to find this equilibrium state; thus, if our assumption is indeed correct, CGS can compute the hidden representation for the equilibrium state and predict the associated features. In fact, CGS employs a variable number of iterations, and the iteration number is defined as the stopping conditions for the steady-state.
>
> Non-implicit implementation of GNN can also find this equilibrium if the number of GNN update is adequately set; for example, the number of iterations is large enough for the model to reach a steady state. However, specifying such proper iteration is often very difficult and, its proper number subject to change depending on a test problem.
>
> **About the multi-head extensions**
>
> The statement of "one layer is enough" from the DEQ paper implies that one equilibrium layer has the same representative power as a serially stacked multiple equilibrium layer. Our multi-head extension of CGS stacks equilibrium layers in a parallel manner so that each of them has a unique fixed point capturing a particular aspect of the target system. This parallelization is intended to increase the expressive power of CGS. The effectiveness of stacking parallel the equilibrium layers is also studied from the direct descendant paper of DEQ [1]; it shows performance gains compared to the single-head DEQ.
>
> **About the runtime comparisons**
>
> |          |      CGS(4)     |      CGS(8)     |     CSG(16)     |     GNN(1)     | GNN(2)          | GNN(3)          | IGNN           | SSE             |
> |:--------:|:---------------:|:---------------:|:---------------:|:--------------:|-----------------|-----------------|----------------|-----------------|
> | Runtimes (sec) | 0.032 $\pm$ 0.004 | 0.051 $\pm$ 0.006 | 0.093 $\pm$ 0.012 | 0.01 $\pm$ 0.001 | 0.018 $\pm$ 0.001 | 0.026 $\pm$ 0.001 | 0.07 $\pm$ 0.008 | 0.261 $\pm$ 0.011 |
>
>
> We provide the runtimes of the CGS and other baseline models when solving physical diffusion problems. By comparing the results of CGS(n) and GNN(1), we can roughly estimate the computation times of the iterative computations. For instance, computing the fixed points of 4 iterative maps took ~ 0.014 seconds (0.032 - 0.018). We can also notice that the computation times of SSE are the highest among the iterative models as they employ the GNN layer itself as the iterative map.
>
> [1] Bai, Shaojie, Vladlen Koltun, and J. Zico Kolter. "Multiscale deep equilibrium models." *arXiv preprint arXiv:2006.08656* (2020).

---

> > ### Comment · Reviewer_R2aV · 2021-11-18
> > **The reply from the reviewer R2aV**
> >
> > Thank you for providing the clarifications and the runtime comparisons that I requested. I keep my score the same (8: accept)

---

### Official Review · Reviewer_G7SD · 2021-11-02

**Correctness:** 3
**Technical Novelty And Significance:** 3
**Empirical Novelty And Significance:** 2
**Recommendation:** 8
**Confidence:** 4

**Main Review:**

The key idea from the observation that the diffusion on a graph often converges to a fixed point. Thus the authors proposed a linear map that is used to mimic this procedure. I did not check all the derivations and proofs of the paper, but I believe the correctness of Theorem 1 is quite obvious.

Currently overall the paper is ok but there are three problem.

(1) The result of Theorem 1 can be further improved. In the paper the authors only consider linear maps, but the Banach fixed point theorem is strong enough to handle nonlinear maps. I guess we can easily add non-linear functions such as ReLU, sigmoid or tanh to (6) to get a nonlinear convergent map.

(2) In linear case, it might not be a good way to compute the fixed point by matrix inverse. The inverse of matrix might be expensive. Meanwhile, by using fast matrix multiplication we can use O(log n) matrix  multiplication to get an approximate fixed point very fast.

(3) In graph value iteration, the authors only considered deterministic transition case, while in practice the transition are often with uncertainty. The authors may consider to extend this part to probabilistic cases.

**Summary Of The Paper:**

The paper constructed a linear map on graph that is guaranteed to converge to a fixed point, by embeding such linear map into a graph neural network, it can be used to solve nonlinear problems on graphs. Experiments show that the proposed methods have good performance in various methods.

**Summary Of The Review:**

The motivation and the theory of the paper is clear and correct. But there are some space for the authors to further improve the quality of the paper, also the value iteration part of the paper is to restrictive and it need to be extended.

---

> ### Author Response · Authors · 2021-11-17
> **Reply for the review G7SD**
>
> Thank you for your thorough inputs on our manuscript. Here, we provide the answer to your review.
>
> **(1)** As you pointed, equation (6) can be extended to consider nonlinear transitions. One possible extension is to apply component-wise non-expansive (CONE) nonlinearities (i.e., each activation functions are non-expansive; ReLU, Tanh) [1] to equation (6). The uniqueness of fixed points can be proven by applying the Banach fixed point theorem. We tested the nonlinear version and shared the results in appendix D.4. We empirically confirmed that having nonlinearity does not give performance gains for the GVI problems. We also like to mention that employing nonlinearity may require more practical care. For instance, the ReLU activation version often produces only 0 fixed points. Bounded activations such as Tanh restrict the values of the fixed points into the region of [-1,1] and underperform the linear version.
>
> **(2)** Sorry for your confusion about the fixed point computation scheme. We calculate the fixed points by using the iterative matrix multiplications. We will update the manuscript so that it will state that we have used iterative matrix multiplications to compute the fixed points. We have also tried to compute the fixed point by directly computing the matrix inversion. Depending on the problem size, the two methods have different computational times. We have provided the computation comparison between these two methods in Figure 8.
>
> **(3)**  We can extend the GVI problem with stochastic transitions. The bellman optimal backup with the consideration of stochastic transitions are given as follows:
>
> $V^{[n]}(s=i)=\max_{j\in \mathcal{N}(i)}(r_{ij}+\alpha\sum_{k \in \mathcal{N}(i)}P(s'=k|s=i, a=j) V^{[n-1]}(s'=k))$
>
> where $P(s'=k|s=i, a=j)$ is the transition probability of transiting from the state $i$ to the state $k$ when the action is decided to transit the state $j$.
>
> We test the CGS and baseline models to solve the stochastic GVI. In the following experiments, we set $P(s'=k|s=i, a=j)=\frac{1}{|\mathcal{N}(i)|}$. We used the same hyperparameters to train all models. The experimental results are given in the **[link](https://drive.google.com/file/d/1ofaDMJHWGJYDr-eHaxZudyFzkXT81An6/view?usp=sharing)**. As shown in the table from the link, CGS still predicts the state values well under the stochastic MDP.
>
> We hope our answer provide enough explanation about your comments. If you have any unresolved questions, please don't hesitate to make additional comments.
>
> [1] El Ghaoui, Laurent, et al. "Implicit deep learning." *SIAM Journal on Mathematics of Data Science* 3.3 (2021): 930-958.

---

> > ### Comment · Reviewer_G7SD · 2021-11-26
> > **Thanks for your rebuttal**
> >
> > Thanks for your rebuttal. Your response covered all my concerns. I will upgrade my score to accept.

---

> ### Comment · Area_Chair_BCDz · 2021-11-26
> **Please respond to the author rebuttal**
>
> Dear Reviewer G7SD,
>
> The authors have posted their rebuttal. I wonder whether the rebuttal addressed your concerns? Please respond to the authors. Thanks!
>
> AC

---

### Official Review · Reviewer_i9Je · 2021-11-03

**Correctness:** 4
**Technical Novelty And Significance:** 3
**Empirical Novelty And Significance:** 3
**Recommendation:** 8
**Confidence:** 5

**Main Review:**

STRONG POINTS: There is a principled theory regarding contractive linear maps as well as the existence of inverses. Experiments are well explained (sometimes is interesting to explain again what is the readout for the classification). Supplementary material is often used by this reviewer just to better understand the statements and experiments, but they are very well structured and provide the necessary evidence.

WEAK POINTS: I get the point of using SSE or IGNN for computing global descriptors of the graphs instead of node embedding, which is not scalable for real-world graphs (large number of nodes). However, the approach presented here is limited exactly for the same reason. Consider for instance the problem of learning many linear mappings for a very large graph. Gradient descent is difficult to compute (need either matrix inversion or solving a linear system) and this is not practical. This is why most of the experiments are done with small graphs.

RECOMMENDATION and ARGUMENTS: My recommendation is weak acceptance. Since the WEAK POINT is evident, the theoretical contribution is very interesting and inspires additional tasks (widely applicable).

QUESTIONS AND ADDITIONAL EVIDENCE. Please clarify the following paragraph: "From the results, we can conclude that learning to generate transition matrices A(G) from the data is more effective than using the fixed transition matrix, as in SSE and IGNN, to achieve a model with better generalizability" and contextualize with respect to the mechanism used by SSE. It there any trade-off between less mappings to learn and better generalization?

**Summary Of The Paper:**

This paper proposes a method for learning how to predict the properties of graphs. These properties are fixed points (e.g. the PageRank vector or Equilibrium Values of a Diffusion Process).Wrt existing methods (SSE and IGNN), CGS learn multiple transition matrices instead of using a unique one. The underlying idea is to: (a) learn several linear operators whose iterations lead to a fixed point, (b) aggregate/combine the fixed points and (c) make the final prediction. With these elements to hand it is possible to find fixed points for several tasks. The proposed approach is tested in PHYSICAL DIFFUSION, GRAPH VALUE ITERATION and GRAPH CLASSIFICATION. The results show that CGSs are competitive with the state of the art and sometimes generalize better.

**Summary Of The Review:**

The paper is theoretically interesting but the results are limited to small graphs.

---

> ### Author Response · Authors · 2021-11-17
> **Reply for the review i9Je**
>
> **About the computational limits of CGS**
>
> We sincerely thank you for your careful review of our manuscript. As you have pointed out from the review, the current formulation and implementation of CGS may not be adequate for the large-sized graphs. However, we would like to mention that the current CGS layer — fixed-point computations — would show the peak memory footprints of roughly $\mathcal{O}(n^2)$ due to the naïve matrix multiplications for computing the fixed points. That memory complexity matches those of the mainstream GNN layers such as GCN, GAT, and GIN.
>
> Furthermore, we can also reformulate the computation procedures of fixed points in distributed settings where the hidden states are updated in a per-node manner. These distributed (or asynchronous) fixed-point calculations also converge to the same fixed points of CGS, similar to where the asynchronous value iterations converge to the real fixed points. Also, those asynchronous updates do not break the end-to-end trainability of CGS since the implicit function theorem does not require the tracking of the gradients along with the internal iterations.
>
> **About the sentence clarification**
> We apologize for our ambiguous writing. The key idea we wanted to address using the sentence `From the results, we can conclude that learning to generate transition matrices A(G) from the data is more effective than using the fixed transition matrix, as in SSE and IGNN, to achieve a model with better generalizability` is that CGS is a more effective solver than other baseline methods (SSE ad IGNN) because CGS constructs the transition matrices $A(g)$ depending on the input graph $g$ while other methods use the constant transition matrices (or model). That is, CGS learns to construct the transition map while considering the problem characteristics and initial conditions captured by the input graph, which is the reason for CGS has a higher generalization capability for solving the various unseen problems.
>
> We will revise this sentence as follows in the revised manuscript:
>
> `The results show that CGS can solve the unseen problems well (i.e., generalization). From the results, we can conclude constructing the transition map adaptively by utilizing the input graph information is a crucial factor for achieving better generalization and higher prediction accuracy of CGS, as compared to other baseline models (SSE and IGNN) that uses fixed transition maps.`
>
> **About the relationship between less mapping and better generalizations**
>
> Using more heads generally increases the expressiveness of CGS. However, the proper number of heads should depend on a target problem. For example, for CGS to learn the PageRank algorithm having the linear updating rule, using a single head is good enough. On the other hand, for CGS to learn the operation of GVI having nonlinearity (i.e., the max operator) parts inside, using more heads is beneficial.

---

> > ### Comment · Reviewer_i9Je · 2021-11-25
> > **Reviewer Response**
> >
> > Thank you for your clarifications, I upgrade my evaluation to accept.
> > Best regards.

---

> > > ### Author Response · Authors · 2021-11-25
> > > **Thank you very much for your response. It seems that the score hasn't yet updated.**
> > >
> > > Sorry to bother you, but it looks like your score is the same as your evaluation score before. I'd really appreciate it if you could check it out and update your score. Thanks again for the very constructive feedback.

---

> > > > ### Comment · Reviewer_i9Je · 2021-11-25
> > > > **Score**
> > > >
> > > > Done

---

> > > > > ### Author Response · Authors · 2021-11-25
> > > > > **Thank you very much!**
> > > > >
> > > > > We really appreciate your re-evaluation.

---

### Author Response · Authors · 2021-11-19
**To all reviewers**

We sincerely thank all reviewers for the careful reviews and inputs. We make some updates to the manuscript so that the revised version can further explain and clarify the issues found during the review period. For the updated parts, we highlight the sentences with blue colors.

The list of updates is as follows:
- Clarifications of experimental/implementation details.
- Explanation of conclusions from the ablation studies.

We also provide the experimental results on the stochastic extension of graph value iteration problems upon the request of the reviewer G7SD. The results can be found from **[this link](https://drive.google.com/file/d/1ofaDMJHWGJYDr-eHaxZudyFzkXT81An6/view)**. In the extended experiments, CGS still shows nice predictive performances.

---

### Decision · Program_Chairs · 2022-01-20

**Decision:**

Accept (Poster)

**Comment:**

The paper got four accepts (after the reviewers changed their scores), all with high confidences. The theories are complete and the experiments are solid. The AC found no reason to overturn reviewers' recommendations. However, the AC deemed that all the pieces are just routine, thus only recommended poster.